

# Integrating SMART principles in Flood Early Warning System Design in the Himalayas

Author: Sudhanshu Dixit[1], Sumit Sen[1*], Tahmina Yasmin[2], Kieran Khamis[2], Debashish Sen[3], Wouter Buytaert[4], David M. Hannah[2]

[1]Centre of Excellence in Disaster Mitigation and Management, Indian Institute of Technology Roorkee, Roorkee, Uttarakhand, India
[2]School of Geography, Earth & Environmental Sciences, University of Birmingham, Birmingham, UK
[3]People's Science Institute, Dehradun, India
[4]Department of Civil and Environmental Engineering, Imperial College London, London, UK
*Correspondence to sumit.sen@hy.iitr.ac.in

Abstract

Extreme precipitation events have increased community and asset vulnerability to hazards like flash floods, particularly in mountainous regions. In response to this challenge, we employ the SMART principle, which emphasizes Inclusiveness and a bottom-up approach, in the development of a comprehensive early warning system for urban floods in lesser Himalayas. A hydrometeorological monitoring network comprising three LiDAR water level sensors and four rain gauges was deployed across the Bindal watershed in Uttarakhand after a meticulous assessment of topography and consultations with local communities. Monitoring reveals that during a monsoon month, a 187 mm difference in rainfall was recorded, with correlations between rainfall at different stations with r = 0.82 down to 0.20 across distances increased from 2.74 to 8.24 km, highlighting significant spatial variability. A southwest movement of rainfall storms, with a 15-minute lag, was observed within the watershed. In contrast to the locally collected data, secondary datasets failed to accurately capture the magnitude and heterogeneity of precipitation patterns, raising concerns about their reliability for flash flood studies at this scale. This study underscores the advantage of SMART approach integrating hydrometeorological insights, utilizing low-cost monitoring systems and community engagement to strengthen urban Himalayan resilience against floods.

Keywords Flood Early Warning System, Flash flood, Himalayas, Rainfall variability



## 1. Introduction

Climate change and water are intrinsically linked, as global warming disrupts the hydrological cycle's balance. It alters rainfall patterns, increases the frequency of extreme rainfall events, and amplifies variability and unpredictability.
(Kulkarni & Agarwal, 2024; Liu et al., 2020; Shi et al., 2021; Upadhyay et al., 2023). Global analysis of daily rainfall records from 1964 to 2013 shows a consistent rise in the intensity and frequency of extreme precipitation events (Papalexiou & Montanari, 2019). These extreme rainfall events often trigger devastating floods and droughts, threatening both lives and infrastructure. A recent global study highlights that 1.81 billion people 23% of the world's population are directly exposed to 1-in-100-year flood events. Most of this at-risk population, around 1.24 billion, live in South and East Asia, with China (395 million) and India (390 million) together accounting for over one-third of global flood exposure (Rentschler et al., 2022). These numbers underscore the critical need for robust Disaster Risk Reduction (DRR) strategies to manage the heightened risks associated with floods.

Flash floods (FFs) are characterized by a rapid and intense surge of water over a localized area (National Weather Service, US), posing unique challenges for mitigation The distinct short time scales and occurrence on small spatial scales differentiate flash floods from other floods, exhibiting rapid onset with minimal lead time for prediction (Collier, 2007; Das et al., 2022). Flash floods rank among the world's deadliest natural disasters (> 5000 deaths annually), constituting approximately 85% of flooding cases and bearing the highest mortality rate among different classes of flooding (Flash Flood Guidance System, WMO). They are driven by several key factors, including intense rainfall that exceeds the ground's absorption capacity, the duration and amount of rainfall, and the spatial-temporal distribution of rainfall, all leading to rapid runoff. In addition to rainfall characteristics, local factors such as urbanization, steep terrain, soil type, and vegetation cover also play a major role (Henao Salgado & Zambrano Nájera, 2022; Hoang & Liou, 2024; Jodar-Abellan et al., 2019; Wang et al., 2023) Rapid urbanization exacerbates the impact, introducing non-stationarity in precipitation extremes through the reduced soil permeability, enhanced convection, and altered impact of large-scale climate circulations (Qian et al., 2022; Song et al., 2022) The urban heat island effect also contributes by raising temperatures in urban areas, further influencing precipitation extremes (Gu et al., 2019; Singh et al., 2016, Gu et al., 2019; Shepherd et al., 2010). Furthermore, there was a significant increase in global urban population growing from 13% in 1900 to 49% in 2005 and is estimated to reach become 60% by 2030 (UN, 2011). Therefore, the climate change along with rapid urbanization intensify the severity and unpredictability of extreme precipitation events, resulting in complex runoff generation mechanisms and devastating flash floods (Lei et al., 2024).

In the Himalayas, the severity of flash floods are exacerbated by steep topography, intense rainfall, rapid snowmelt, and fragile ecosystem presenting considerable threat to life and property (Rasmussen & Houze, 2012; Satendra et al., 2015). States along the southern Indian Himalayas, such as Himachal Pradesh, Sikkim, and Uttarakhand, are experiencing growing risks from flash floods, mainly driven by intense and erratic rainfall (Dimri et al., 2016). The interplay between evolving climatic conditions and rapid urbanization exacerbates these risks, emphasizing the necessity for enhanced forecasting and mitigation strategies to safeguard vulnerable populations in the Himalayan region (Arlikatti et al., 2018). Developing effective flood Early Warning Systems (EWS) is essential for addressing the vulnerability of Himalayan watersheds to flash floods. EWS is a critical component of the Disaster Risk Reduction (DRR) framework, which aims to minimize the impact of natural hazards and enhance preparedness. Various approaches have been proposed to detect and predict FFs with increased accuracy and lead time (Bloschl et al., 2008; Hapuarachchi et al., 2011; N. Zhou et al., 2024), including conventional flood monitoring systems with limited range and sophistication (Zakaria & Jabbar, 2021a). However, major sources of uncertainty in Flood Early Warning Systems (EWS) are due to:

1. Lack of high-resolution reliable rainfall observations, as they are crucial in accurate flood predictions but are scarce in high altitude regions like the Himalayas (Chawla & Mujumdar, 2020; Gebrechorkos et al., 2024);

2. Challenges with secondary datasets - Satellite and reanalysis data face resolution and accuracy issues due to high altitude and complex steep terrain (Dogra et al., 2023);
3. Limited knowledge of hydrological processes - Insufficient knowledge of hydrological dynamics in complex Himalayan watersheds results in inaccuracy of predictive models (Nanda et al., 2018; Nepal et al., 2014)
4 Lack of inclusiveness - When the local community is not involved, it weakens local resilience and reduces the ability of the system to address diverse socio-economic vulnerabilities.
Addressing these challenges is crucial for enhancing the reliability and functionality of Flood EWS (Ward et al., 2011; Yasmin et al., 2023; Zakaria & Jabbar, 2021b).
Therefore, the present work emphasizes adopting SMART approach (Shared understanding of risks, Monitoring of risks, building Awareness, Response action on Time) proposed by Yasmin et al. (2023), which promotes inclusiveness



and a bottom-up strategy to maximize local relevance and effectiveness. Community involvement strengthens EWS by integrating local knowledge into system design and decision-making. Unlike traditional statistical models that rely on fixed thresholds, a community-driven approach allows for the dynamic determination and regular adjustment of warning thresholds based on changing river channel conditions. This makes flood warnings better suited for the local conditions, flexible, and ultimately more reliable for decision making.

The present study, building on the principles of the SMART approach and the dynamical thresholding, takes a step towards enhancing flood early warning systems (EWS) in urban, mountainous Himalayan regions, where data availability is limited and watershed characteristics exhibit high variability. The primary objectives of this work are: 1) to demonstrate the benefits of low-cost, real-time monitoring systems and the importance of fostering community engagement in flood risk management; and 2) to integrate the spatiotemporal variability and dynamic nature of

watershed characteristics into flood early warning systems. We hope this study's findings will contribute to the development of more adaptive and context-specific flood forecasting approaches, serving as an essential step towards more effective and localized flood risk management.

## 2.   Study Area

The study focuses on a mountainous watershed of Bindal river in the Western Himalayan state of Uttarakhand. The

Bindal River (78°1'6.087"E, 30°16'17.699"N) in the Lesser Himalayas is a relatively small watershed covering 44.4 km². Originating from the base of the Mussoorie ridges, the Bindal River is sustained by numerous springs and headwater streams. Serving as a tributary to the Ganga River, it traverses through Dehradun, the capital of Uttarakhand. It is characterized by rugged terrain, with a substantial elevation ranging from 539 m to 997 m (Figure 1a).

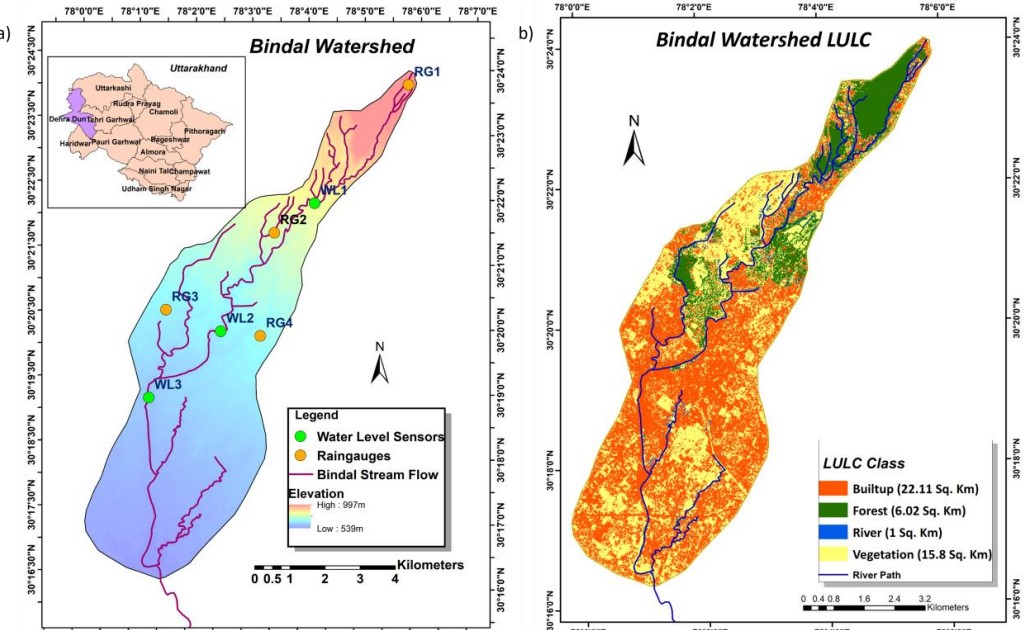

*Figure 1 Figure 1: Bindal Watershed. a) Map of Bindal watershed with sensor locations, i.e., rain gauges: RG1, RG2, RG3, RG4, and river water level recorders: WL1, WL2, and WL3;  b) Land Use/Land Cover (LULC) of Bindal watershed.*

The Bindal River is ephemeral in nature and remains dry or experiences low flow throughout the year. However, the probability of flash floods increases during the monsoon months or when sudden atmospheric depressions cause short bursts of heavy rainfall (V. V. et al., 2016). Riverside settlements are the house residents with limited incomes and

restricted earning prospects. Due to the infrequent flow, riverine communities often perceive these rivers as less





hazardous. Consequently, over time, they encroach upon the riverbanks, gradually reducing the river's width and exposing human settlements to potential hazards (Bansal et al., 2015; Habeeb & Javaid, 2019).

### 3. Methodology

The methodology of the present work is divided into four key components, encompassing various stages that emphasize inclusiveness, a bottom-up approach, and attention to hydrological dynamics. These stages range from initial engagement with stakeholders (local authorities and affected communities) to the analysis of collected hydrological data and the development of an efficient early warning system (Figure 2). The first step involved interacting with local communities to understand their perception of flooding, identify flood-prone areas along rivers, and build trust in the early warning system. Another crucial aspect was determining flood level thresholds based on the observed impact of water levels on the community. The second step involves real-time monitoring and data collection. Telemetry sensors are installed across the watershed to measure rainfall and river water levels, with their placement guided by community input and watershed topography. In addition, satellite and reanalysis data are collected for comparative analysis. The third step focuses on understanding the watershed by conducting land use/land cover analysis and assessing rainfall variability to gain deeper insights into the study area. This includes comparing observed data with secondary sources to evaluate their reliability for integration into the early warning system. Finally, the fourth step aims to characterize rainfall patterns and watershed dynamics to establish a robust and efficient early warning system.

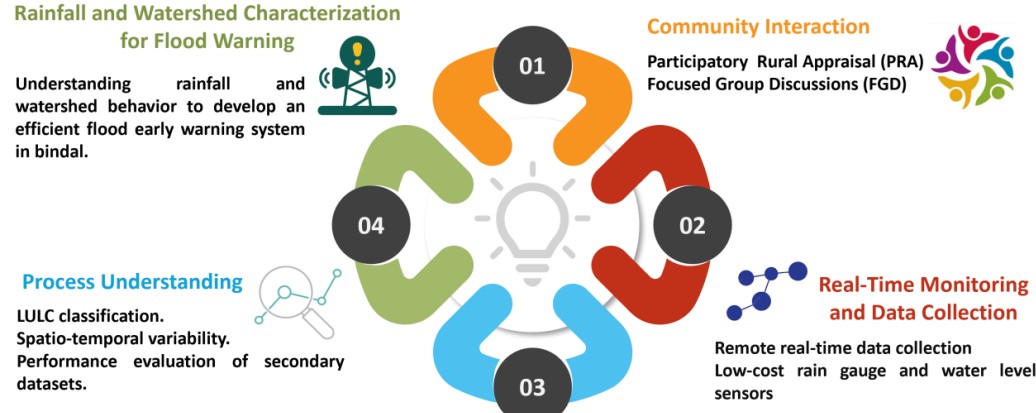

Figure 2 The key components of the methodology adopted in the present work.

### 3.1. Community Interaction

Effective communication with the community is essential for implementing an early warning system. This involves understanding the community's perspective on flooding and its impacts. It helps enhance our understanding of context-specific risks, vulnerabilities, and exposure levels. Moreover, involving the community fosters a sense of ownership and builds trust in the flood EWS. For the above purpose, the Bindal river was divided into four different stretches (i) Starting from Kanwali Road Bridge to Laal Bridge covering an area of 2 kms on both the banks of river (ii) Laal Bridge to Bindal Bridge (Haridwar Bypass) of 2.7 Km, (iii) Kali Mandir (Kargi Patel Nagar Bypass Road) Bridge to Lohia Nagar, and (iv) Bindal Bridge (Haridwar Bypass) to Chandchak Bridge which covers an area of 3.0 km on the right bank and 2.7 km on the left bank. Community interactions were held with the different resident colonies along both banks of the river as per the above stretches. The communities in these colonies mainly include households who have migrated and settled from Bihar, Uttar Pradesh, and other parts of Uttarakhand itself.

Participatory Rural Appraisal (PRA) exercises (Reddy et al., 2016), along with Focus Group Discussions (FGDs) with concerned community members from the above colonies, including slums along the four mentioned stretches, were conducted to gather crucial information required to develop early warning systems for the Bindal watershed. The PRA



exercises included historical transect, analyzing trends, mapping resources and making seasonal and Venn diagrams. Approximately 100 households participated in these exercises. Historical transects in addition to trend analysis helps to examine the shifts in base flows, frequency and severity of floods, settlement density and impacts of flooding, including erosion, infrastructure damage, and livelihood loss. Social resource mapping was undertaken to explore encroachment around the river, the built-in infrastructure in the different stretches, and further identify the vulnerable

sites and most affected households based on past floods. and the reasons behind them. Season diagramming helped in understanding the seasonal and peak flows in the Bindal river as per the residents. Community representatives also prepared Venn Diagrams indicating the different stakeholders related to river management and discussions were held around their role and effectiveness. These interactions ended with inviting suggestions for a flood early warning system.

A stratified random sample survey of 100 affected households was further conducted in the critical locations identified in different stretches, considering their sources of livelihoods (Services, Business, Labour, and Unemployed). These surveys helped in identifying the most affected individuals and how flooding impacts them. Engagement with the communities also allowed us to determine the optimal locations for recording water level and establishing rain gauges.

### 3.2. Real-Time Monitoring and Data Collection

In-Situ Sensor Data Collection

Real-time data collection used cost-effective telemetry rainfall and water level sensors strategically deployed to comprehensively understand rainfall dynamics and precisely capture variability within the Bindal River watershed. Lidar-based distance sensors were used for water level measurement. Lidar ranging operates by utilizing the roughness

of the reflective surface to produce nonspecular reflection, or scattering, of the laser beam. The application of Lidar for distance measurement is promising due to its cost-effectiveness, high energy efficiency, and minimal spatial footprint. The sensor can take measurements up to an incidence angle of approximately 40° with an accuracy of around 1 cm for distances less than 10 m. If also achieves a maximum detectable range of 30–35 m (Paul et al., 2020). The Davis tipping bucket rain gauge can measure rainfall levels ranging from 0 to 10.2 cm/hr, with a resolution of 0.2 mm

and an opening diameter of 16.5 cm, covering a collection area of 214 cm² (Lanza et al., 2022).

Three Lidar-based water level sensors (WL1, WL2, WL3) were installed along the riverine length, and four tipping bucket rain gauges (RG1, RG2, RG3, RG4) were installed across the entire watershed at optimal locations (Figure 1a). Water level sensors WL1, WL2, and WL3 were located at the watershed's upper, middle, and lower sections, respectively. Rain gauges are strategically positioned in the Bindal watershed to provide comprehensive coverage of

rainfall variability across altitudinal zones. R1, situated at the highest point, captures rainfall patterns in mountainous terrain, while R2, R3, and R4, positioned successively lower, provide data as rainfall moves between foothills and plains. This configuration thoroughly explains the rainfall distribution and intensity gradients in the watershed. These telemetry water level sensors capture data every 5 minutes, while the telemetry rain gauges record rainfall at 15-minute intervals. Data collection commenced in September 2022. For seasonal analysis, rainfall data was divided into

monsoon (JJAS) and non-monsoon periods (ONDJFMAM). Three distinct periods—late monsoon 2022, non-monsoon 2023, and monsoon 2023—were analyzed based on data availability.

Secondary Data Acquisition:

Sentinel-2 images of the study area from the European Space Agency Copernicus program were utilized for Level 1 Land Use/Land Cover (LULC) classification (Isbaex & Margarida Coelho, 2021). Optical data captured in September

2022 were obtained (Figure 1b), offering high spatial resolution (HR) imagery with a cloud cover of less than 0.08%. GPM (IMERG) Version 06— satellite data and ERA5 — the global atmospheric reanalysis data were utilized to evaluate the effectiveness of secondary rainfall data in capturing heterogeneity.(Hersbach et al., 2020; Skofronick-Jackson et al., 2017)

### 3.3. Process Understanding

LULC Classification



Variations in land cover control the surface runoff mechanism and provide a better understanding of the study area. As the built-up area decreases, infiltration increases and surface runoff decreases, whereas an increase in forest area enhances infiltration and reduces surface runoff. Sentinel-2 images of 10 m spatial resolution were classified using ArcGIS 10.5's Maximum Likelihood (ML) supervised classification tool. Training data and spectral signature files were created to represent four uniform classes (Built-up, Forest, River, and Vegetation (including agriculture, scrubland, grassland)). All LULC maps produced using this technique have an accuracy of more than 80% (Ahmad & Quegan, 2013; Ali et al., 2019).

Spatio-temporal variability

Spatiotemporal variability analysis is critical in hilly terrain due to elevation, complex topography, and microclimate interplay. Rapid elevation changes create localized weather patterns, while mountains can cause rain shadows and orographic effects. Therefore, rainfall and water level observations were analyzed to unveil spatial and temporal variability and characterize the watershed response. The observed rainfall as a function of distance, monthly cumulative rainfall observed at each rain gauge, and intraday variability during different periods capture spatial and temporal rainfall variability in the Bindal watershed. It is important to note that rain gauges with partial data availability are excluded from the comparison for that month.

Performance Evaluation of Secondary Datasets

In assessing the accuracy and capability of secondary datasets to capture rainfall dynamics, various descriptive indices (Table 1) were employed, including Percentage Bias (PBIAS), Root Mean Squared Error (RMSE), and cross-correlation. PBIAS gauges the average tendency of simulated datasets to overestimate or underestimate the reference dataset. By evaluating the dispersion of predicted values around observed data points, RMSE provides a concise measure of data performance. Cross-correlation between two timeseries measures the degree and nature of the relationship between them as they evolve. The value ranges between [-1,1]. By examining cross-correlation, potential dependencies and lead-lag relationships between variables can be identified, providing valuable insights into temporal associations and patterns. (Prakash, 2019; Prakash et al., 2012).

Table 1 Formulae used for various metrics.

| Descriptive Indices | Formulae |
|---|---|
| Percent bias (PBIAS) | $\dfrac{\sum_{i=1}^{n}\left(Y_i^a - Y_i^b\right) \times 100}{\sum_{i=1}^{n}\left(Y_i^a\right)}$ |
| Root mean square error (RMSE) | $\sqrt{\dfrac{1}{n}\sum_{i=1}^{n}(Y^a - Y^b)^2}$ |
| Cross-Correlation (CC) | $\dfrac{\sum_{i=1}^{n}(Y_i^a - \overline{Y}^a)\left(Y_i^b - \overline{Y}^b\right)}{\sqrt{\sum_{i=1}^{n}(Y_i^a - \overline{Y}^a)^2} \times \sqrt{\sum_{i=1}^{n}\left(Y_i^b - \overline{Y}^b\right)^2}}$ |

Where $Y_i$, $\overline{Y}$, represents the rainfall, mean rainfall, and $Y^a$, $Y^b$, $\overline{Y}^a$, $\overline{Y}^b$ represent rainfall and mean rainfall of two different time series.

### 3.4. Rainfall and Watershed Characterization for Flood Warning

The watershed response to extreme rainfall patterns is a crucial factor to provide accurate and reliable flood warnings (Knighton & Walter, 2016). Therefore, the present section emphasizes understanding the association between extreme



rainfall events and changes in the water level of the Bindal River. The observed water levels recorded within the watershed in response to different extreme rainfall events were classified into five types of alerts based on water level thresholds and corresponding actionable warning (Table2), using a threshold-based approach adapted from (Young et al., 2021). Water level thresholds are determined through a combination of statistical analysis and community involvement. Flood thresholds are determined by analyzing observed water level data to isolate extreme events, typically defined as values exceeding the 99th percentile. This percentile-based approach provides an objective criterion for identifying rare, high-impact flood occurrences. In addition to this, community engagement was

undertaken to validate and refine the flood thresholds, integrating local knowledge and past experiences of flood events obtained through direct interaction with community members. Given the dynamic geomorphology of the narrow Bindal stream driven by anthropogenic factors such as waste material dumping, encroachment, and channel modification, a purely statistical approach to defining flood thresholds may be inadequate. These ongoing changes necessitate the integration of contextual and community-informed insights for more accurate flood characterization.

Table 2 Five levels of flood alerts, their thresholds, and the action required.

| Type of alert | Threshold | Action |
|---|---|---|
| Warning | 99.99 percentile of Water level | Flood-like situation: Evacuate. |
| Advisory | 99.9 percentile of Water level | Flood-like situation: Stay away from banks |
| Watch | 99.5 percentile of Water level | Stay alert |
| Information statement | 99 percentile of Water level | No action required |
| Cancellation | Below 99 percentile of Water level | Safety confirmed |

A rainfall event is defined as a continuous period of rain lasting at least 30 minutes and any observations with minimum rainfall magnitude exceeding 2mm. By leveraging finer spatial and temporal resolutions, critical watershed dynamics, including rainfall storm movement direction, watershed response patterns, and rainfall trends, were studied. These

insights are invaluable for developing an effective early warning system.

To understand the severity of the rainfall, the daily rainfall has been classified into different categories based on its magnitude (Table 3). The India Meteorological Department (IMD) a national agency responsible for meteorological observations and weather forecasting, has established a classification system for rainfall severity based on its magnitude (Bhatla et al., 2019).


Table 3 Classification of daily rainfall based on IMD (India Meteorological Department) criteria. (source: Bhatla et al., 2019 )

| Rainfall classification | Daily rainfall(mm) |
|---|---|
| No rainfall | 0 mm |
| Very light rainfall | 0-2.4 mm |
| Light rainfall | 2.5-7.5 mm |
| Moderate rainfall | 7.6-35.5 mm |
| Rather heavy rainfall | 35.6-64.4 mm |
| Heavy rainfall | 64.5-124.5 mm |
| Very heavy rainfall | 124.6-244.4 |





## 4. Results and Discussion

### 4.1. Community involvement

Participatory Rural Appraisal (PRA) activities and Focus Group Discussions (FGDs) conducted with the local communities in the study area helped in gaining insights into the nature of floods. Below listed are the key findings from these exercises.

- The base flow of the river has decreased in the last 40 years. The water level in the river increases in the rainy season (July-Sept) up to 10-15 feet, but it remains as low as 0.5-1 feet throughout the year.
- The frequency and intensity of floods have been increasing over the last two decades, especially in the past 3-4 years attributed to intensified rainfall events. According to the locals, in the last 15 years there had been three major floods (2006, 2013 & 2018) with devastating impact.
- The human settlement along the banks of the Bindal River has increased since the 1970s. Encroachment on the riverbed was reported in all the stretches, reducing the width of the main course, which was identified as
one of the major reasons for high flood intensity.
- In the stretch between Bindal Bridge and Chanchak Bridge (Figure 3), a lot of illegal riverbed mining activity is carried out throughout the year (except during the rainy season) near Shiv Colony and Muslim Colony on the left bank.
- Localities near the riverbanks remain at extreme risk during the monsoon. Damage caused by flooding was
mostly in the form of bank erosion and breakdown of retaining walls, affecting public and private property.
- Retaining walls at many places (Gandhi Gram, Sanjay Colony, Sangam Vihar, Sattowali Ghati, Dhera Khas, Kabadi Market, Muslim Colony and Chandchak) get damaged frequently during rainfall. One of the many high-tension wire towers constructed along the Bindal River from Kanwali Road Bridge to Laal Bridge stretch collapsed in the year 2018.

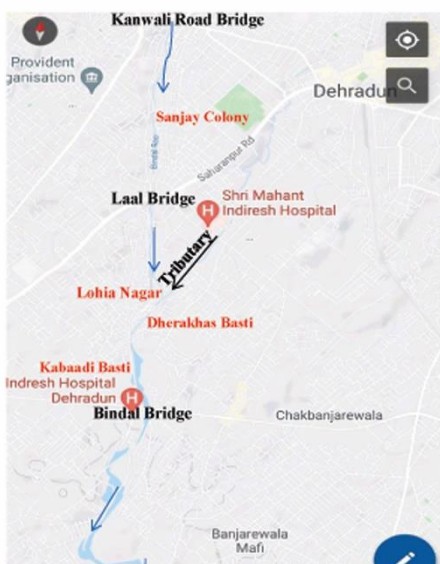

*Figure 3 Map showing the course of Bindal river. Key locations are marked for reference. Source: Google map(as on 25th August 2024) © Google Maps.*

The above findings highlight certain areas where the extent and impact of flooding are significantly greater than in other localities. Out of the 21 colonies surveyed (8 on the left bank and 13 on the right bank), five colonies—Sanjay Colony and Dhera Khas Basti on the left bank, and Lohia Nagar, Mehboob Colony (also known as Kabadi Basti), and Chota Bharuwala on the right bank were identified as the most vulnerable. These flood-prone areas, identified through PRA and FGDs, are useful for selecting appropriate locations for water level monitoring installations.

Another key observation is that the factors contributing to floods in the Bindal River area align with findings from previous studies. As noted by (Gaur et al., 2020), Bindal functions as a stormwater channel, but its condition is deteriorating primarily due to garbage dumping, especially plastics, which significantly reduce the flow capacity of the drainage channels. Similarly, (Bansal et al., 2015a) highlighted that river flooding is largely driven by garbage dumping and encroachments while flooding in other parts of the city is due to inefficient stormwater drainage. These findings suggest that relying solely on conventional hydrological or hydraulic models with static thresholds for flood forecasting may not be effective. This is because such thresholds are highly sensitive to local conditions, such as the accumulation of debris in river channels. Therefore, thresholds should be determined at the




local level and updated regularly in response to changes in channel morphology. Overall, these insights reinforce the importance of localized flood risk assessment and adaptive threshold-setting to improve the effectiveness of early warning systems in the Bindal River context.

### 4.2. Spatio-temporal variability

The spatial variability of rainfall is illustrated through a cumulative monthly rainfall analysis, as depicted in Figure 4. Months with cumulative rainfall greater than 200 mm are shown. Plots for all the months can be found in the Supplementary Information (figure S1). Figure 4 shows significant variations in cumulative rainfall, showcasing distinct rainfall patterns observed within the watershed. During the late monsoon period in September 2022, there was a significant 187 mm difference between RG1 (higher elevation) and RG4 (lower elevation). Additionally, during the non-monsoon period from January 2023 to May 2023, cumulative rainfall differences across the rain gauges exceeded 200 mm. Higher elevations witness substantial total rainfall, which gradually decreases towards lower elevations. This trend persists for 5 out of 10 months, from September 2022 to June 2023, with RG1 showing higher rainfall, as seen in the supplementary Figure S1. These findings are coherent with existing literature. (Henn et al., 2018a; Nogueira, 2020; Shrestha et al., 2012) and can be attributed to orographic rainfall, where mountain barriers influence airflow and augment regional rainfall over lower hills (Kitchen & Blackall, 1992; Prat & Barros, 2010).

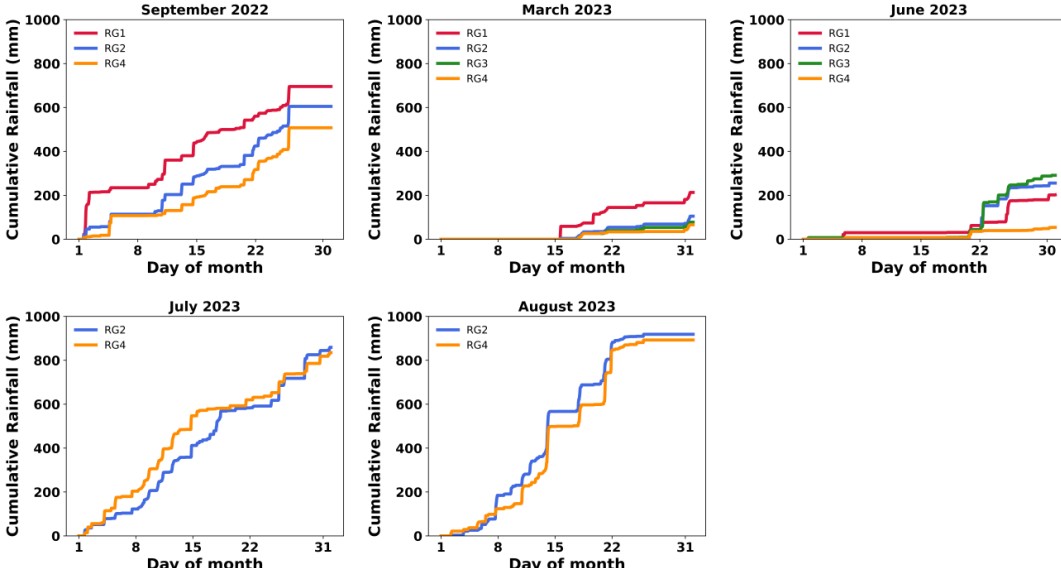

Figure 4 Cumulative plot of monthly observed rainfall across different rain gauges in the watershed (only months with more than 200 mm of cumulative rainfall are shown), illustrating spatial variability in rainfall. Higher elevations generally receive more rainfall than lower elevations. Additional plots for all months (September 2022–August 2023) are provided in the Supplementary Information (Figure S1).

The spatial variability of rainfall at the seasonal scale is illustrated in Figure 5, which presents a correlation analysis across different distances. The correlation coefficient was computed between pairs of rain gauges for each season and plotted against the distance (in kilometers) separating them. During the non-monsoon period, cross-correlation, representing the similarity between rain gauge time series, generally declined from 0.83 to 0.24 over distances ranging from 2.7 to 8.24 km. A similar trend was observed during the monsoon seasons of 2022 and 2023, where the correlation dropped from 0.55 (2022) and 0.70 (2023) to below 0.1 and 0.26, respectively, over the same distance. This indicates a sharper decrease in similarity during the monsoon season. The steeper slopes observed during the monsoon periods (0.086 and 0.081) suggest more rapid spatial rainfall variability than the non-monsoon period. Although it is well established consistent with Tobler's law that similarity in rainfall decreases with increasing distance (Pavlopoulos &



Gupta, 2003), it is particularly noteworthy that significant variability is evident even across relatively short distances of up to 8 km within this small watershed.

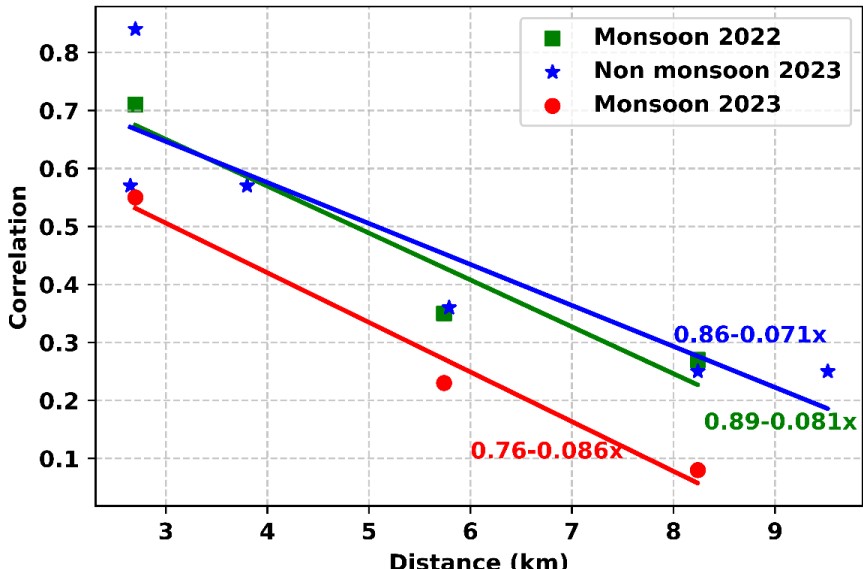

Figure 5: Correlation between rainfall observed at different rain gauges as a function of distance, analyzed at a seasonal scale. The plot highlights the rapid decline in similarity (high variability) over short distances within the watershed, particularly during the monsoon season.

The above analysis spotlights the pronounced spatial heterogeneity of rainfall within a relatively compact watershed, shaped by both topographic influences and seasonal dynamics. The cumulative rainfall differences, consistently observed across multiple months, and the sharp decline in inter-gauge correlation over short distances highlight the limitations of relying on sparse or single-point rainfall measurements in complex terrains. These findings emphasize the importance of high-resolution spatial rainfall monitoring, particularly for hydrological modeling, flood forecasting, and water resource management in mountainous or orographically influenced regions (Buytaert et al., 2006; Z. Li et al., 2014; Panziera et al., 2015) Ultimately, traditional methods relying on sparse gauge networks or uniform rainfall assumptions may lead to underestimation of flood hazards in such terrains. Integrating spatially distributed hydrological models that account for elevation-driven rainfall variability, along with real-time radar data and probabilistic forecasting techniques, can significantly enhance the accuracy and reliability of EWS in mountainous watersheds (Rafieeinasab et al., 2015).

### 4.3.    Secondary Data Comparison

The high spatiotemporal rainfall variability observed within a small watershed highlights the necessity of employing high-resolution data to effectively capture such variability for hydrological modeling and understanding watershed dynamics. When a high density of observed data points is unavailable, alternative options, such as secondary datasets, become necessary. Therefore, their performance evaluation becomes an important task in obtaining reliable results.




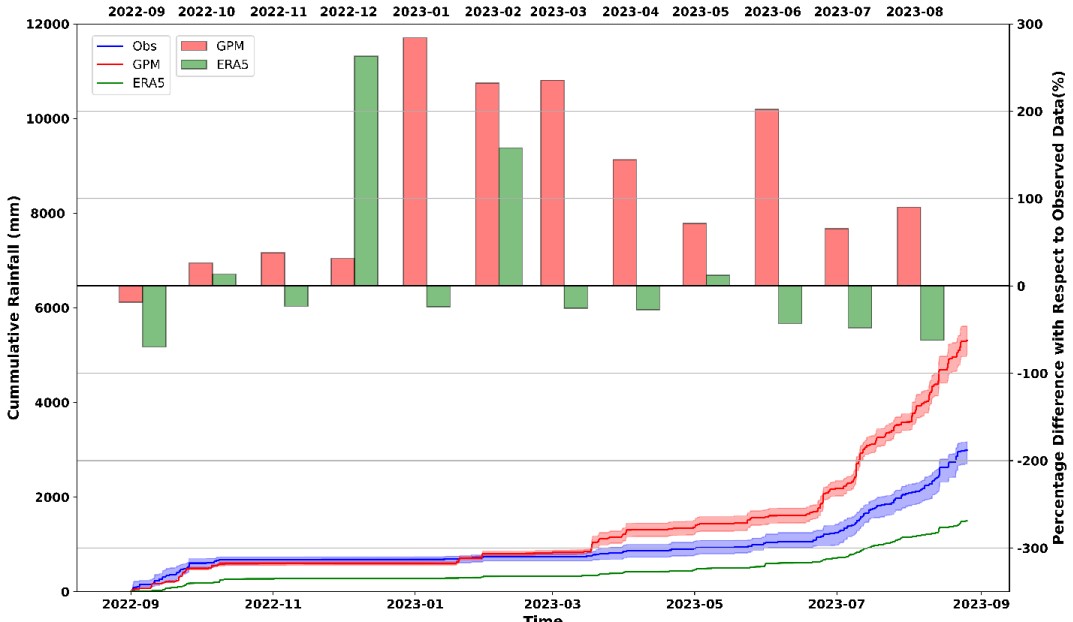

Figure 6 Comparison of cumulative rainfall measurements over time for different datasets (Obs, GPM, ERA5) and their percentage difference with respect to observed data.

Figure 6 compares cumulative rainfall between observed and secondary datasets from late monsoon 2022 to monsoon
2023. Further, it illustrates the monthly percentage difference in cumulative rainfall between the observed data and
the secondary data. The ERA5 dataset has a resolution of 0.25° × 0.25°, covering the watershed area within a single
pixel. In contrast, the GPM data utilizes a higher resolution of 0.1° × 0.1°, resulting in two pixels covering the
watershed. The observed data is derived from four rain gauges. The graph displays ERA5, GPM, and observed data,
with spatial variability represented by shaded areas.

Upon comparing the secondary datasets, it becomes evident that neither the satellite nor the reanalysis adequately
represents the rainfall of this watershed. However, during the monsoon period, ERA5 underestimates rainfall, with
discrepancies exceeding 40% compared to observed monthly cumulative rainfall. Additionally, it demonstrates mixed
behavior in the non-monsoon period, with significant percentage differences (>100%), particularly when cumulative
monthly rainfall is low (<50 mm). These findings underscore ERA5's underestimation of rainfall throughout the year.
Furthermore. For example, the percentage difference was around 200% in June, exceeding 70% on average.

Table 4 Statistical comparison of secondary rainfall (GPM-IMERG and ERA5) with observed rainfall

|                                 | GPM-IMERG | ERA5    |
| ------------------------------- | --------- | ------- |
| PBIAS                           | -43.57    | 100.16  |
| RMSE (Hourly rainfall)          | 3.65      | 2.24    |
| CC (Hourly rainfall)            | 0.117     | 0.173   |
| RMSE (Daily Rainfall)           | 32.95     | 18.36   |
| CC (Daily rainfall)             | 0.488     | 0.456   |
| Cumulative annual Rainfall (mm) | 5306.53   | 1495.8  |

When compared to using descriptive indices in Table 4, ERA5 outperformed GPM in RMSE, with lower values of
2.24mm (hourly) and 18.36 mm (daily) compared to 3.65mm and 32.95mm, respectively. ERA5 also exhibited a
higher correlation coefficient (0.173 for hourly rainfall) than GPM (0.117). However, GPM showed a higher



correlation for daily rainfall. Despite these differences, both datasets displayed biases—GPM overestimated rainfall while ERA5 underestimated it. The cumulative observed rainfall was 2994 mm. ERA5 shows less error and better captures the hourly pattern, while GPM effectively captures the daily rainfall pattern. Overall, ERA5 tends to overestimate light rainfall and underestimate moderate to heavy rainfall, while GPM exhibits the opposite behavior, tending to underestimate light rainfall and overestimate moderate to heavy rainfall (Chen et al., 2023; Xin et al., 2021).

Secondary rainfall data, while valuable, faces limitations in accurately capturing the heterogeneity of rainfall patterns, especially in regions with complex terrains and urban areas. It often fails to represent heterogeneity at higher temporal and spatial resolutions, as evidenced by correlation coefficients of 0.117 and 0.173 (Henn et al., 2018b; Z. Zhou et al., 2019). Therefore, there is an urgent need for high-resolution, long-term rainfall datasets having spatial and temporal variability. This is critical to analyze the interplay between rainfall variability, watershed heterogeneity, and hydrological response in complex terrains and urban areas (Cristiano et al., 2017).

### 4.4. Watershed Dynamics and Warning

Rainfall and water level data to understand the watershed dynamics and its dependence on flood warnings. The peak rainfall intensity at 15-minute interval was determined at each rain gauge across three distinct periods: a) Monsoon 2022, b) Non-monsoon 2023, and c) Monsoon 2023.

Figure 7 illustrates the probability density function (PDF) of the 15-minute maximum rainfall intensity observed at each rain gauge across different seasons. The Y-axis represents density, a smoothened estimate of the probability distribution of the data.

During the monsoon season, distinct patterns emerge in peak intensities across various rain gauges, while non-monsoon periods lack such clear trends. Within the monsoon season, there is a gradual increase in the magnitude of maximum intensity downstream, particularly when examining rare events such as the tail of the distribution or extreme values of density. In monsoon 2022, RG1 shows higher density at lower rainfall intensity (20-30 mm/hr), while RG2 and RG4 display more varied distributions with notable peaks around 30-50 mm/hr. The same trend of rising peak intensity magnitudes from RG1 to RG4 is observed in late monsoon 2023. For instance, RG1 shows a 15-minute maximum rainfall density tail in the range of 70-80 mm/hr, which increases downstream to a range of 120-140 mm/hr at RG3 and RG4. In contrast, during the non-monsoon period, mixed behavior is observed among RG2, RG3, and RG4, while RG1 shows the highest peak intensity, opposite to the behavior observed during the monsoon period. The observed rainfall patterns are in congruence with the existing literature, indicating less intense but more sustained rainfall in forest areas (Siyum, 2020), while in urban areas, more intense and shorter-duration rainfall events are observed (Yang et al., 2024). This observation suggests that urban areas located at lower elevations are more susceptible to experiencing higher-intensity events (Y. Li et al., 2020).



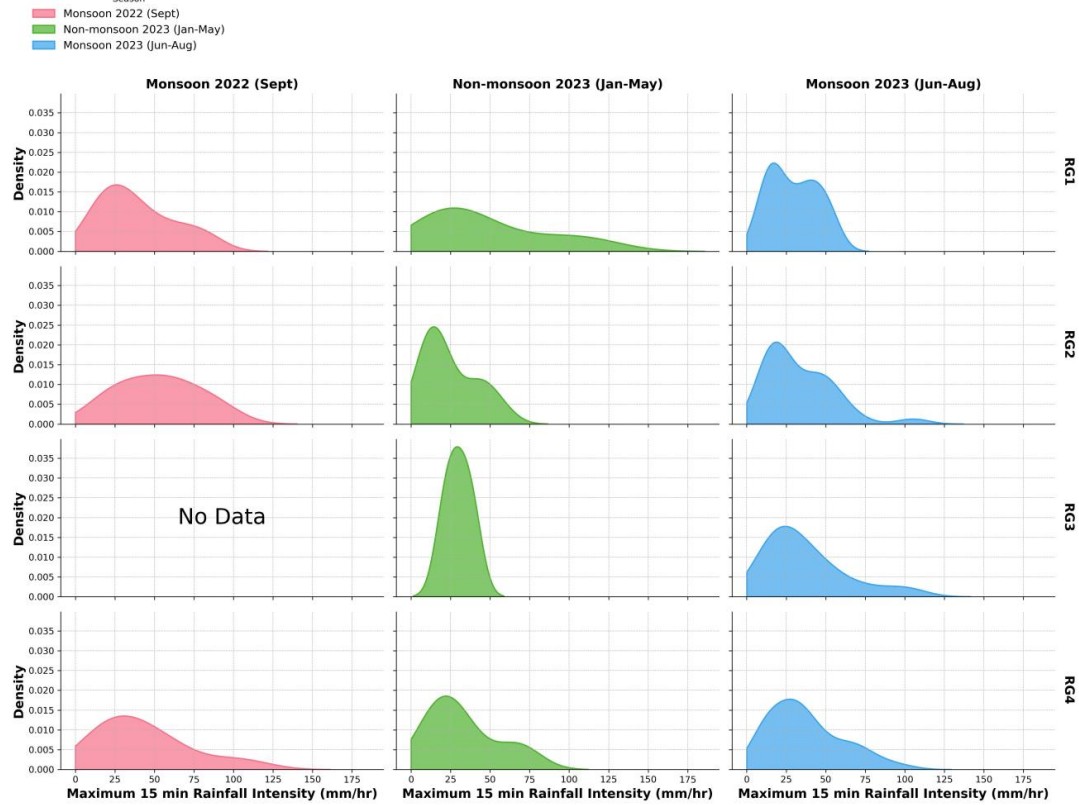

Figure 7: Density plots showing the distribution of maximum 15-minute rainfall intensities (mm/h) observed at various rain gauges across different seasons within the watershed. The plots highlight distinct seasonal patterns, with notable differences between monsoon and non-monsoon periods, and variations in intensity magnitudes across gauges and downstream locations.

In addition to peak rainfall intensities, understanding the rainfall storm's direction is crucial.to improve the lead time for early warning. Correlation between the rain gauges can be used to characterize the rainfall storm's movement within the watershed, rainfall data from all four rain gauges, observed at 15-minute intervals, were correlated with each other at varying lags from 0 to 5, as tabulated in Table 5. The maximum correlation was found at lag 1, equivalent to a 15-minute delay. Notably, there was a significant increase in correlation between RG1 and RG3/RG4 at lag 1, indicating a stronger association between these rain gauges during rainfall events. Our observations showed a consistent southwest movement of rainfall storms originating from RG1 (point at the highest elevation) and progressing towards RG3, where urban settlements are predominant at lower elevations. This southwest movement is likely to be caused by the combined effects of orography, prevailing wind patterns, and the urban heat island. The orographic effect leads to increased rainfall at higher elevations, while prevailing winds may cause the downward movement of rainfall storms. Further, the urban heat island effect can alter local atmospheric conditions and airflow patterns, influencing the spatial distribution of rainfall within the watershed (Wu et al., 2021). These mechanisms contribute to the observed downward movement of rainfall storms within the watershed, impacting rainfall patterns and intensity in urban areas at the lower elevation. The interaction between storm motion and a basin drainage network directly affects flood peaks. For example, storms moving downstream (aligned with river flow) generate faster hydrological responses and higher peak discharges compared to upstream-moving storms (Amengual et al., 2021). This occurs because downstream-moving storms synchronize rainfall intensity with water accumulation in channels.



Table 5 Lagged correlation between observed rainfall of rainguages within the watershed. Shaded rows indicate where the correlation initially increases with lag and then decreases.

| Lag 0 (No Delay) | Lag 1 (15-min Delay) | Lag 2 (30-min Delay) | Lag 3 (45-min Delay) | Lag 4 (60-min Delay) | Correlated Rainguage | Lagged Rainguage |
|---|---|---|---|---|---|---|
| 0.278 | 0.248 | 0.154 | 0.102 | 0.070 | RG1-RG2 | RG2 |
| 0.109 | 0.129 | 0.120 | 0.088 | 0.063 | RG1-RG3 | RG3 |
| 0.189 | 0.190 | 0.094 | 0.055 | 0.044 | RG1-RG4 | RG4 |
| 0.278 | 0.196 | 0.184 | 0.106 | 0.071 | RG2-RG1 | RG1 |
| 0.447 | 0.495 | 0.401 | 0.350 | 0.273 | RG2-RG3 | RG3 |
| 0.561 | 0.454 | 0.203 | 0.140 | 0.084 | RG2-RG4 | RG4 |
| 0.109 | 0.087 | 0.083 | 0.070 | 0.065 | RG3-RG1 | RG1 |
| 0.447 | 0.309 | 0.200 | 0.140 | 0.121 | RG3-RG2 | RG2 |
| 0.343 | 0.186 | 0.131 | 0.119 | 0.085 | RG3-RG4 | RG4 |
| 0.189 | 0.112 | 0.095 | 0.062 | 0.051 | RG4-RG1 | RG1 |
| 0.561 | 0.371 | 0.206 | 0.110 | 0.067 | RG4-RG2 | RG2 |
| 0.343 | 0.360 | 0.258 | 0.152 | 0.132 | RG4-RG3 | RG3 |

The observed rainfall at all the rain gauges has been categorized according to the IMD criteria (Table 3), and the frequency of each rainfall category is presented in Table 6 below. The missing data was assumed to have zero rainfall and hence categorized as 'No Rainfall.' There were 21 days with heavy rainfall and 5 days classified as very heavy rainfall, which are recognized as potential causes of flash floods.

Table 6 Classification of rainfall based on IMD criteria. The table displays the number of days associated with different categories of rainfall.

| | No rainfall | Very light | Light | Moderate | Rather heavy | Heavy | Very heavy |
|---|---|---|---|---|---|---|---|
| RG1 | 243 | 40 | 39 | 20 | 12 | 4 | 1 |
| RG2 | 231 | 26 | 50 | 18 | 24 | 8 | 2 |
| RG3 | 288 | 17 | 29 | 14 | 8 | 3 | - |
| RG4 | 244 | 19 | 53 | 22 | 13 | 6 | 2 |

Understanding the response time of the watershed is crucial for effective flood warning systems. Utilizing a lagged correlation method and comparing the time difference between peak rainfall and peak water level, it was determined that the watershed exhibits a range of response times from 15 minutes to 2 hours and 30 minutes. Specifically, this response time varies from 15 to 45 minutes during heavy and very heavy rainfall events. Out of the 26 days with heavy/very heavy rainfall observed at any rain gauge, four days were selected to represent the complex watershed



dynamics in response to the rainfall, as illustrated in Figure 8. A high degree of rainfall distribution was observed,
with some rain gauges recording heavy/very heavy rainfall while others recorded very light, light, and moderate
rainfall. Further, it was observed that even for heavy/very heavy rainfall, flood warnings ranged from watch and
advisory to warning levels, highlighting the role of complex watershed dynamics in addition to rainfall variability in
predicting the severity of the flood.

Although Figure 8(a-d) is plotted for heavy/very heavy rainfall categories, strikingly different responses from the
watershed are observed. In Figure 9a, three rainfall events of different duration and magnitude occurred, but the
response in water level is almost the same. When examining the rainfall of all the rain gauges for 25/09/2022 and
21/08/2023 (Figure 9b, 9d, and Supplementary), a similar pattern in rainfall categories is evident. All are in heavy and
very heavy categories; on 25/09/2022, 73, 89, and 92.4 mm of rainfall were recorded in RG1, RG2, and RG4,
respectively, while on 21/08/2023, 106.8, 167.6, and 242.2 mm of rainfall were recorded in RG1, RG2, and RG4.
Despite being in the heavy/very heavy category, the magnitude of rainfall on 21/08/2023 was more severe. However,
in contrast, the water level at WL3 downstream increased more on 25/09/2022, crossing the warning threshold, which
is greater than the water level of WL3 on 21/08/2023, which only crossed the advisory threshold.

Further, it is interesting to note that on July 17th, 2023, RG2 and RG3 recorded heavy rainfall with magnitudes of
76.8 mm and 124.2 mm, respectively, while RG1 and RG4 simultaneously reported very light and light rainfall with
amounts of 0.4 mm and 3.2 mm, respectively. This variation in rainfall distribution across the watershed resulted in a
different response compared to the other three cases, with only a watch alert being issued. This underscores the
importance of considering localized rainfall patterns and their impact on watershed dynamics when issuing flood
alerts.

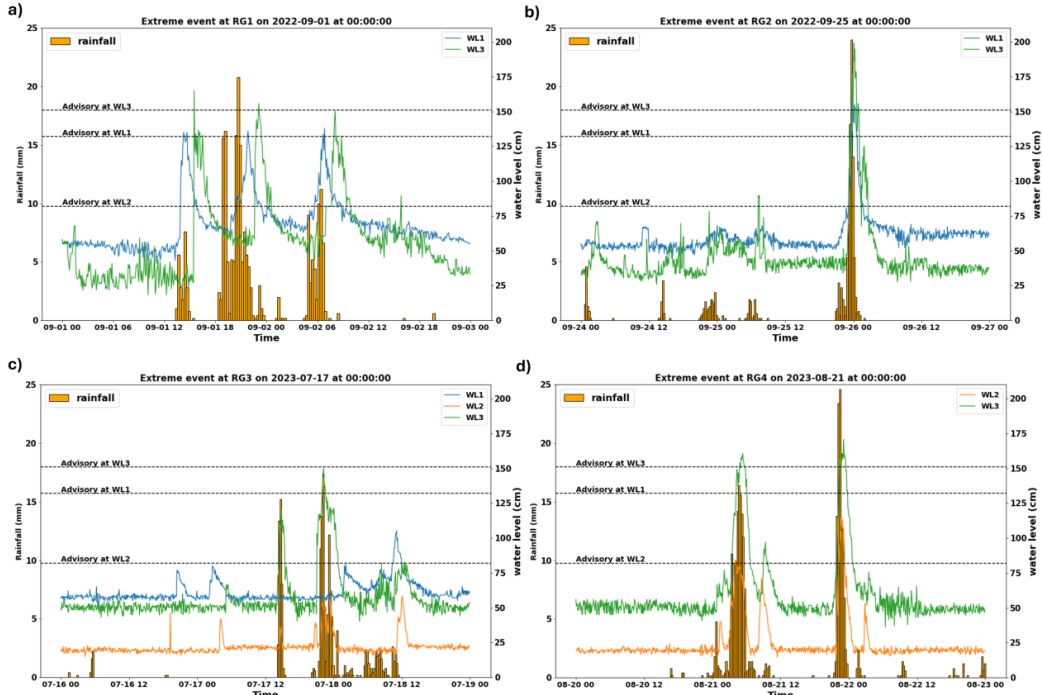

Figure 8 The response of water level sensors WL1, WL2, and WL3 to extreme rainfall events in the watershed, highlighting the
variability in watershed dynamics. Advisory thresholds for water level alerts are indicated for each site (black dashed line). Panels
a) to d) correspond to extreme rainfall events on different dates, showing the water level response and rainfall data from RG1 to
RG4, respectively.



### 5. Conclusion


This study underscores the potential contribution of SMART approach of integrating low-cost, real-time watershed monitoring system with community-centric strategies to improve FEWS. Particularly in data-scarce, complex urban Himalayan settings of Bindal watershed, the inclusiveness and bottom-up approach of the SMART principle show the importance of community involvement in building an integrated approach to disaster risk reduction. Community

participation helps gather on-ground information regarding the severity of floods, foster trust in the EWS, and identify vulnerable areas. This engagement also fostered a sense of responsibility and ownership, ensuring the optimal location and maintenance of sensor network. The community input was essential in defining flood-level thresholds as statistical methods alone may not suffice in dynamic watersheds like the Bindal, where human-induced changes alter the channel characteristics over time. Thus, Integration of community-driven insights provide a responsive and context-specific

approach to efficient FEWS. Rainfall and water level data from four rain gauges and three water level sensors were analyzed for characterizing the spatiotemporal variability of rainfall, storm movement, and watershed dynamics.

The detailed examination of extreme rainfall events sheds light on the watershed response to intense rainfall, revealing significant spatial and temporal variability in rainfall patterns within the 44.4 km² watershed. Higher elevations receive more rainfall, with a notable difference of 187mm observed in monthly rainfall between rain gauges situated 8.28 km

apart, while lower elevations experience higher-intensity rainfall compared to higher elevations. Additionally, the work compares the performance of secondary data with observed rainfall using descriptive metrics such as PBIAS, RMSE, and correlation coefficient, revealing inadequacies in both satellite and reanalysis data representation of rainfall in this watershed. ERA5 underestimates, while GPM data overestimates rainfall, emphasizing the need for a high-resolution monitoring network for reliable flood (EWS) in regions with complex terrains and urban areas. Our

results also indicate that rainfall storms tend to move southwest downstream, with a lag of approximately 15 minutes observed between RG1 and RG3. The urban mountainous region exhibits complex hydrological behavior, where flood occurrences are influenced not only by meteorological factors, such as upstream rainfall patterns, but also by watershed-specific characteristics. Therefore, relying solely on heavy or very heavy rainfall at a single gauging point may be insufficient for accurate flood prediction and management in such terrains. These findings shows the critical

importance of analyzing both rainfall patterns and watershed responses to extreme events for enhancing flood Early Warning Systems (EWS). Notably, the study demonstrates that small watersheds can exhibit rapid responses even to minor rainfall inputs. This highlights the need to consider watershed-scale variability and dynamics in hydrological assessments to support more effective flood management strategies.

Watersheds are inherently complex systems, where interactions among topography, land cover, and rainfall govern

water flow and flood behavior. A comprehensive understanding of these dynamics, including rainfall patterns and runoff generation, enables better anticipation of and response to flash flood events. Such insights support the identification of vulnerable areas, the development of accurate flood models, and the timely issuance of warnings, ultimately strengthening community safety and resilience. Thus, integrating low-cost monitoring systems with hydrometeorological analysis and community engagement is essential for developing holistic early warning systems.

These systems must not only be data-driven and responsive to watershed dynamics but also inclusive of community knowledge and participation.

### 6. Data availability

The rainfall and water level datasets collected through field instrumentation are not publicly available but can be obtained from the corresponding author upon request.



7. Author contributions

SD: Study conception and design, literature review, data collection, analysis and interpretation of results, and draft manuscript preparation. SS, KK, and DMH: Contributed to study conception and design, and provided feedback to refine the interpretation of results. All other authors: Reviewed the results, provided feedback to improve interpretation, and contributed to drafting the manuscript.

8. Competing interests

The authors declare that they have no conflict of interest.

9. Acknowledgements

The authors sincerely thank the Natural Environment Research Council (NERC) for financial support under grant no. NERC COP26 A&R, Project Scoping Call-2021COPA&R31Hannah. We are also grateful to the Centre of Excellence in Disaster Mitigation and Management (CoEDMM), IIT Roorkee, for providing essential resources, facilities, and support that were instrumental to this research. The first author gratefully acknowledges the support of the Prime Minister's Research Fellowship (PMRF), Grant No. 2802448. We also extend our special thanks to Prof. Srikrishnan from CoEDMM for his valuable discussions and insightful input on the manuscript.

10. Financial support

This research has been supported by the Natural Environment Research Council (grant no. NERC COP26 A&R, Project Scoping Call-2021COPA&R31Hannah).

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
