# Peer review of "Integrating SMART principles in Flood Early Warning System Design in the Himalayas"

_EGUsphere, 2025_

## Community Comment (CC1)

**EGUsphere-2025-2081 Review comment:**

**Title of Manuscript:**

Integrating SMART principles in Flood Early Warning System Design in the Himalayas

**Reviewer Comments and Suggestions for the Authors:**

The paper addresses critical societal and scientific problems which is important to enhance flood early warning system. The authors have conducted a thorough review of related documents, and considering the local community's inclusiveness and participation through various techniques, as well as the involvement of stakeholders in the current study, is vital. However, there are many editorial corrections; the authors should review the manuscript to improve the coherence, consistency, and the overall paper quality. The following comments and questions are provided to the authors to improve the final document.

**Abstract:**

- Line 15: we employ the SMART principle… It is important to define abbreviations when first used. What is SMART? Please also consider the same comments for all abbreviations and acronyms as well.

- Line 19: Monitoring reveals that during a monsoon month, a 187 mm difference in rainfall… this sentence is not clear, rewrite the whole statement. Is this 187 mm difference, seasonal or monthly record? I think there is monsoon season not monsoon month.

- Line 23: secondary datasets failed to accurately capture the magnitude and heterogeneity of precipitation patterns,… what are the secondary datasets? It is good to disclose them if possible.

**Introduction:**

Improve this section to make it more consistent and coherent.

Line 61: Furthermore, there was a significant increase in global urban… you should use a more recent estimated data…(UN, 2011) is too old for such information.

Line 74: …to detect and predict FFs … Define abbreviations when first used FFs? And once defined use the short form consistently throughout the document e.g., Flood Early Warning Systems (EWS) used multiple time, such as Line 71, 76 and 96 correct it.

Line 100: We hope this study's findings will contribute… better if this sentence is modified in such way: The findings of this study is anticipated to contribute…

Line 110: *Figure 1 Figure 1:*

Based on Figure 1, the LULC map, it is understandable that the Watershed is urban watershed. Have you considered the effect of urban drainages systems, interactions of urban solid waste on the flood generations?

What is the source and ground resolution of DEM used in this study? As the watershed area is smaller the effect of DEM resolution used for topography analysis in such studies are important.

Line 143: Chandchak Bridge which covers an area of 3.0 km on the right bank and 2.7 km on the left bank….is this area or length/width? Or convert the Unit into km.sq.

Have you used a standard formula or procedure to select sample size of the survey 100? It may be important to indicate it in the manuscript how you decided to select 100 affected participants from the total population of the study area.

Line 155: What is Season diagramming? Is it seasonal analysis of the flood peak? So many new terms.

Line 173: If also achieves a maximum… Correction: It also achieves a maximum.

It is important to indicate the rain gauges and water levels recorders' detailed locational information, preferably in Table X and Y coordinates with altitude and period of data records as well.

Line 211: Performance Evaluation of Secondary Datasets, is it right to say secondary datasets or Global climate datasets? Please refer to use the right term. Remote sensing data are not secondary data.

Table 2 Five levels of flood alerts, their thresholds, and the action required. The Threshold used to define flood alert are very high and narrow ranges as indicated in Table 2. Can you visualize the difference in the flood threshold of 99.99 vs 99.9? Are this threshold economically feasible? Please also review related papers and field experience on this Table 2.

Figure 3 can be presented in a more improved way, with gauging stations, urban colonies and other important map features included in it.

**Spatio-temporal variability**

For depicting the spatial variability of rainfall over the catchment, it is also important to produce the spatial change in the form of map map-based spatial surface or preferably interpolated map with suitable techniques in addition to Figure 4.

Line 333: The above analysis, which part of the above? Please refer to the Figure or Table.

Based on Table 4 Statistical comparison of secondary rainfall (GPM-IMERG and GPM) with observed rainfall, it shows the annual rainfall (mm) GPM 5306.53, and GPM 1495.8 mm significantly varied. Have you made a bias correction or downscaling analysis? It is important to bias correct global databases before application.

Line 383 to Line # 387, the statements are not clear, please re-write them.

In general leaf shaped or elongated watersheds generate less flood peak compared to oval or circular shaped watersheds, other factor kept constant. The current or Bindal watershed is more or less leaf shaped, I also assume that is why the peak flows a bit lagged the peak rainfall event evident from Figure 8. Can you please discuss this issues more based on the result obtained in the current study?

---

## Author Comment (AC1)

**RESPONSE TO COMMENTS (CC1)**

**Integrating SMART principles in Flood Early Warning System Design in the Himalayas (NHESS-2025-2081)**

**Authors: Sudhanshu Dixit[1], Sumit Sen[1*], Tahmina Yasmin[2], Kieran Khamis[2], Debashish Sen[3], Wouter Buytaert[4], David M. Hannah[2]**

[1]Centre of Excellence in Disaster Mitigation and Management, Indian Institute of Technology Roorkee, Roorkee, Uttarakhand, India

[2]School of Geography, Earth & Environmental Sciences, University of Birmingham, Birmingham, UK

[3]People's Science Institute, Dehradun, India

[4]Department of Civil and Environmental Engineering, Imperial College London, London, UK

*Correspondence to sumit.sen@hy.iitr.ac.in

**Dear Editor and Reviewers,**

The authors would like to thank the reviewer for their careful review of our manuscript and for providing valuable comments and suggestions, which we found very helpful in improving the manuscript's quality. We have carefully addressed all your comments and integrated your insightful suggestions into the revised manuscript. In the subsequent detailed response, we have addressed each comment individually. Comments are written in red, and our responses follow each comment in black. All the new details added in the manuscript are highlighted as text in italics. For your reference, the sources cited in our responses can be found in the references section on the last page of this document. We look forward to your positive feedback and hope you will find the revised manuscript satisfactory.

1. The paper addresses critical societal and scientific problems which is important to enhance flood early warning system. The authors have conducted a thorough review of related documents and considering the local community's inclusiveness and participation through various techniques, as well as the involvement of stakeholders in the current study, is vital. However, there are many editorial corrections; the authors should review the manuscript to improve coherence, consistency, and overall paper quality.

Thank you for your careful consideration and thorough evaluation of our manuscript. We sincerely appreciate your encouragement and motivation for this article. The authors agree that the manuscript can be improved in light of coherence, consistency, and overall paper quality. Please see the detailed responses below for each of your comments.

2. Line 15: we employ the SMART principle... It is essential to define abbreviations when they are first introduced. What is SMART? Please also consider the same comments for all abbreviations and acronyms as well.

Thank you for your careful examination. *SMART refers to **S**hared understanding of risks, **M**onitoring of risks, building **A**wareness, **R**esponse action on **T**ime.* For brevity, we have used the acronym in the abstract, as it should be limited to 200 words. However, in the main manuscript (Introduction Line 88) the acronym has been explained thoroughly before being used in the subsequent sections

3. Line 19: Monitoring reveals that during a monsoon month, a 187 mm difference in rainfall... this sentence is not clear, rewrite the whole statement. Is this 187 mm difference, seasonal or monthly record? I think there is monsoon season not monsoon month.

Thanks for pointing out the lack of clarity in the sentence. Here, the authors indicate that the two stations recorded a total rainfall difference of 187 mm during the monsoon month of September 2022. We have revised lines 19-21 as follows:

"*Monitoring reveals that during the monsoon month of September 2022, a 187 mm difference in rainfall was recorded, with correlations between rainfall at different stations with r = 0.82 down to 0.20 across distances increased from 2.74 to 8.24 km, highlighting significant spatial variability.*"

4. Line 23: secondary datasets failed to accurately capture the magnitude and heterogeneity of precipitation patterns. what are the secondary datasets? It is good to disclose them if possible.

Thanks. The secondary datasets used in this study are GPM IMERG (Global Precipitation Measurement Integrated Multi-Satellite Retrievals for GPM) and ERA5. We have revised lines 22–24 as follows:

'*In contrast to the locally collected data, secondary datasets (GPM IMERG, ERA5) failed to accurately capture the magnitude and heterogeneity of precipitation patterns, raising concerns about their reliability for flash flood studies at this scale.*'

5. Line 61: Furthermore, there was a significant increase in global urban... you should use a more recent estimated data...(UN, 2011) is too old for such information.
   Thank you for your careful examination. We have modified the statement according to the latest available data and referenced sources. The revised manuscript now reads: *'Furthermore, there was a significant increase in the global urban population, with the proportion rising from 30% in 1950 to 55% in 2018, and it is projected to increase to 68% by 2050 (UN, 2019).'*

6. Line 74: ...to detect and predict FFs ... Define abbreviations when first used FFs? And once defined use the short form consistently throughout the document e.g., Flood Early Warning Systems (EWS) used multiple time, such as Line 71, 76 and 96 correct it.

Thank you for the valuable comment. We have defined all abbreviations at their first occurrence (e.g., Flash Floods (FFs) and Early Warning Systems (EWS)) and have ensured consistent use of the short forms throughout the manuscript. The revisions have been made at Lines 71, 76, and 96 accordingly.

7. Line 100: We hope this study's findings will contribute... better if this sentence is modified in such way: The findings of this study is anticipated to contribute...

Thank you for your insightful suggestion regarding line 100. We have modified the sentence to improve clarity and formality. The revised sentence now reads:

*"The findings of our study are anticipated to develop adaptive and context-specific flood forecasting approaches, serving as an essential step towards effective and localized flood risk management."*

8. Line 110: Figure 1 Figure 1:  Based on Figure 1, the LULC map, it is understandable that the Watershed is urban watershed. Have you considered the effect of urban drainages systems, interactions of urban solid waste on the flood generations? What is the source and ground resolution of DEM used in this study? As the watershed area is smaller the effect of DEM resolution used for topography analysis in such studies are important

Thank you for your observation regarding urban drainage systems and the role of urban solid waste in flood generation. In the present study, these aspects have not been explicitly considered because our focus is on developing a community-integrated, data-driven early warning system rather than a physically or process-based model. Our approach is centered on integrating community inputs and real-time data streams for flood warning

We appreciate your comment on the importance of DEM resolution, particularly for small watershed studies where topographic details are critical. In our study, the ALOS PALSAR digital elevation model (DEM) is used, which provides a spatial resolution of 12.5 meters.

9. Line 143: Chandchak Bridge which covers an area of 3.0 km on the right bank and 2.7 km on the left bank....is this area or length/width? Or convert the Unit into km.sq.

Thank you for your observation. We have revised lines 137-144, as follows.

*Effective communication with the community is essential for implementing an early warning system. This involves understanding the community's perspective on flooding and its impacts. It helps enhance our understanding of context specific risks, vulnerabilities, and exposure levels. Moreover, the community fosters a sense of ownership and builds trust in the EWS for floods. As per the transect survey conducted by the research team and discussions held with local stakeholders, the lower reach of the Bindal River is more affected by floods as compared to its upper and middle reaches. Therefore, the lower reach of the Bindal river was further divided into three different stretches (i) from Kanwali Road Bridge to Laal Bridge – 1.82 km, (ii) from Laal Bridge to Bindal Bridge (located at Haridwar Bypass) - 2.11 km, and (iii) from Bindal Bridge (located at Haridwar Bypass) to Chandchak Bridge – 2.28 km. In addition to the above, another stretch covering a small tributary of Bindal i.e. from Kali Mandir (located at Kargi Patel Nagar Bypass Road) Bridge to Lohia Nagar –2.40 km, was also considered. For detailed community interactions, an area of 4.21 sq. km. was considered covering all the flood prone areas of neighborhood situated on both the banks of the river, along the above stretches. These neighborhoods come under 12 wards of Dehradun city. Community consultations were further held with the different neighborhood residents along both banks of the river as per the above stretches. The communities in these neighborhood mainly include households who have migrated and settled from Bihar, Uttar Pradesh, and other parts of Uttarakhand.*

10. Have you used a standard formula or procedure to select sample size of the survey 100? It may be important to indicate it in the manuscript how you decided to select 100 affected participants from the total population of the study area.

Thank you for bringing the concern regarding the sampling procedure and size to our attention. The section (Lines 160-163) has been rewritten to clarify the concerns raised.

*According to the PRAs and FGDs conducted with the residents of neighborhoods located along the four selected stretches of the Bindal river, it was found that the severity of flood damages was much more in the second stretch (Laal Bridge to Bindal Bridge) as compared to third stretch (Bindal Bridge to Chandchak Bridge) and first stretch (Kanwali Road Bridge to Laal Bridge), in decreasing order. The neighborhoods along the fourth stretch were reported to be the least affected. A qualitative case-study research methodology was adopted, for which the five most vulnerable localities were further identified along the three stretches during the participatory resource mapping exercises. In each of these five localities, 20 households were selected in consultation with the ward members, considering differences in their sources of livelihoods (Services, Business, Labour, and Unemployed) to understand the disparity of impacts during floods, if any. This resulted in a total sample size of 100 households for the case study of Bindal floods. These surveys helped identify the most affected households and understand the impact of flooding on them. It revealed that households having more unemployed members and income primarily from labour, living closer to the riverbanks, suffered the most through loss of household assets and needed more recovery time.*

11. Line 155: What is Season diagramming? Is it seasonal analysis of the flood peak? So many new terms.

Thank you for highlighting the terminology confusion. We confirm that 'season diagramming' refers to the analysis of monthly variation in river discharge, including flood peaks, for the four different river stretches considered for the study. Specifically, it involved a participatory assessment of flood magnitude across different seasons.

12. Line 173: If also achieves a maximum... Correction: It also achieves a maximum.

 Thank you for pointing out this grammatical error. We have corrected the sentence in the revised manuscript to "*It also achieves a maximum*".

13. It is important to indicate the rain gauges and water levels recorders' detailed locational information, preferably in Table X and Y coordinates with altitude and period of data records as well.

Thank you for your valuable recommendation. In response, we have added detailed tables (ST2) in the supplementary materials, which provide the latitude and longitude coordinates for all rain gauges and water level recorders used in this study. For completeness and ease of reference, we have also included these tables below in the response document. Please note that water level sensors were installed on April 1, 2022, and rain gauges on September 1, 2022. Data collection from these sites has been continuous via telemetry, with interruptions only occurring when sensors malfunctioned. The data collection promptly resumed following maintenance and repair, resulting in minimal gaps in the observational record.

| Sensor | Name | Latitudes | longitudes |
|---|---|---|---|
| Rainguage1 | RG1 | 30.39833333 | 78.09541667 |
| Rainguage2 | RG2 | 30.35973333 | 78.05485 |
| Rainguage3 | RG3 | 30.34092778 | 78.02178889 |
| Rainguage4 | RG4 | 30.33598611 | 78.04876389 |
| Waterlevel1 | WL1 | 30.335517 | 78.038947 |
| Waterlevel2 | WL2 | 30.36703333 | 78.06797222 |
| Waterlevel3 | WL3 | 30.31888889 | 78.01607222 |

14. Line 211: Performance Evaluation of Secondary Datasets, is it right to say secondary datasets or Global climate datasets? Please refer to use the right term. Remote sensing data are not secondary data.

Thank you for raising this important point regarding dataset classification. Sorry for any confusion caused. We use the term "secondary datasets" here collectively to refer to data that the authors did not collect firsthand but obtained from external sources, including satellite remote sensing and global climate products. This terminology is also used in

research, such as Venkatesh et al. (2020), who explicitly refer to satellite and reanalysis precipitation products as secondary datasets when applied in hydrological evaluation and modelling.

We understand that terminology can vary and appreciate the importance of clarity. However, in our study, since these datasets are externally sourced rather than primary data collected via field observations, we consider the term "secondary datasets" appropriate and consistent with several usages in hydrologic and climate science literature.

15. Figure 3 can be presented in a more improved way, with gauging stations, urban colonies and other important map features included in it.

Thank you for your valuable suggestion to enhance Figure 3 by including additional map features. We agree that these improvements enhance the clarity and informativeness of the figure by providing better spatial context for the study area. Accordingly, we have revised Figure 3 by incorporating the requested features and ensuring that all map elements are clearly labeled and visually distinguishable. We have repositioned the figure

and placed it below Section 3.1, which provides clearer reference to the locations discussed in the community interaction section. This placement will also support Section 3.2 by helping readers better understand the positions of the rain gauge and water-level monitoring stations. Additionally, we have added a new Figure 3b in Section 4.1, which provides a zoomed-in view of the updated figure for improved detail and interpretability.

[Figure]

The updated figures are available in the revised manuscript.

16. Table 2 Five levels of flood alerts, their thresholds, and the action required. The Threshold used to define flood alert are very high and narrow ranges as indicated in Table 2. Can you visualize the difference in the flood threshold of 99.99 vs 99.9? Are this threshold economically feasible? Please also review related papers and field experience on this Table 2.

Thank you for your insightful comment regarding the flood alert thresholds detailed in Table 2. The table is summarized as follows:

| Type of alert | Threshold | Action |
|---|---|---|
| Warning | 99.99 percentile of Water level | Flood-like situation: Evacuate. |
| Advisory | 99.9 percentile of Water level | Flood-like situation: Stay away from banks |
| Watch | 99.5 percentile of Water level | Stay alert |
| Information statement | 99 percentile of Water level | No action required |

| Cancellation | Below 99 percentile of Water level | Safety confirmed |

In our study, thresholds were determined using a data-driven approach consisting of two components: (1) statistical analysis to identify extremes, and (2) community-based assessment and validation. We utilized a long-term, high-resolution water-level dataset recorded at five-minute intervals from April 2022 to May 2024, comprising more than 200,000 data points from a single monitoring station. This extensive dataset enabled us to capture a wide range of hydrological conditions, including major flood events. The statistically derived thresholds were subsequently reviewed and validated by local community members, who contributed their historical knowledge and recent experiences to classify appropriate actions and assess the likelihood of risks to property and life. An important advantage of this statistical approach is that the thresholds will continue to become more robust as additional data are collected over time, ultimately supporting more precise and reliable early warnings.

Although the numerical difference between the 99.99 and 99.9 percentiles may appear small, the large number of observations means that the associated exceedance events are substantial, with more than 20 water level observations for one threshold and over 200 for the other in some cases. These differences translate into significant variations in flood magnitude, which depend on the local river cross-sectional width at sensor locations. And, as data collection continues, these observed datasets will grow, enabling the thresholds to become more robust over time.

For example, at the sensor WL3 site, where the river width is approximately 26 m, the difference in water level between the 99.9 and 99.99 percentile thresholds is 0.27 m. At a narrower site, WL1, where the width is ~12 m, this difference increases to about 1.0 m. In the context of this narrow, small river system, such differences are meaningful and clearly distinguish hazard levels.

17. Spatio-temporal variability

For depicting the spatial variability of rainfall over the catchment, it is also important to produce the spatial change in the form of map-based spatial surface or preferably interpolated map with suitable techniques in addition to Figure 4.

Thank you for this valuable suggestion. We have incorporated monthly spatial rainfall distribution maps generated using Inverse Distance Weighting (IDW) interpolation for the Bindal watershed. These new maps show rainfall heterogeneity across the catchment for each month from September 2022 to August 2023, with active rain gauges for each respective period. IDW is a widely used spatial interpolation technique that estimates continuous rainfall surfaces from discrete measurement points, providing clearer insight into localized rainfall patterns.

[Figure]

*Figure: Monthly cumulative rainfall spatial variation across the Bindal watershed from September 2022 to August 2023. Rainfall distribution is generated using Inverse Distance Weighting (IDW) interpolation with a power parameter p=2. The maps display rainfall values (mm) along with the locations of the active rain gauges used for interpolation during each respective month.*

However, a key limitation of spatially interpolated maps, such as IDW, is that they represent cumulative rainfall totals over a given month and thus cannot capture temporal variation or variability within individual rainfall events during that period. Unlike monthly cumulative rainfall line plots, which effectively visualize the number of rainfall events, their magnitudes, and spatial differences during each event, spatial maps provide only a static view of rainfall distribution for the entire month. These factors limit the spatial plot to reflect dynamic intra-month rainfall characteristics, such as intensity and timing, and frequency of event changes.

Therefore, to maintain focus and clarity, these spatial rainfall maps are included in the supplementary materials rather than the main manuscript, complementing the temporal rainfall analysis presented in Figure 4.

18. Line 333: The above analysis, which part of the above? Please refer to the Figure or Table.

Thank you for pointing out the need for clarity. We have revised the sentence to explicitly refer to the analysis presented in Figures 4 and 5 to guide the reader clearly to the relevant visual data supporting the discussion. The revised manuscript now states: "*The above analysis shown in Figures 4 and 5...*"

19. Based on Table 4 Statistical comparison of secondary rainfall (GPM-IMERG and GPM) with observed rainfall, it shows the annual rainfall (mm) GPM 5306.53, and GPM 1495.8 mm significantly varied. Have you made a bias correction or downscaling analysis? It is essential to bias-correct global databases before application.

Thank you for highlighting the importance of bias correction and downscaling in the use of global rainfall datasets. We acknowledge that bias correction and downscaling are crucial steps in enhancing the applicability of global datasets for local hydrological modeling and applications. *In this study, we did not perform explicit bias correction or downscaling prior to comparison, as our objective was to assess the raw, inherent performance and biases of two readily available, high-temporal-resolution secondary rainfall products (GPM-IMERG and ERA5) relative to ground-observed rainfall. The rationale for this approach was to determine whether these global products, due to their accessibility and high temporal resolution, could adequately capture the variability and magnitude of precipitation within the smaller, topographically complex semi-urban Himalayan watershed.* The results indicate that neither raw product sufficiently captures the observed precipitation magnitude and variability (as discussed in Section 4.3), highlighting the necessity of a sensor network (rain gauges) to capture the hydrometeorological dynamics of the Himalayan watershed.

We will include this justification in Section 3.3 of the revised manuscript to clarify the rationale behind this approach.

20. Line 383 to Line # 387, the statements are not clear, please re-write them.

Thank you for your valuable comment regarding the clarity of the statements in lines 383 to 387. We agree that the original text could be clearer. Accordingly, we have rewritten this section to improve readability and provide a more precise explanation of the analysis and figure presentation. The revised text now reads as follows:

"*Rainfall and water level data were analyzed to understand watershed dynamics and their relationship with flood warnings. The peak rainfall intensity at 15-minute intervals was calculated for each rain gauge during three distinct periods: (a) Monsoon 2022, (b) Non-monsoon 2023, and (c) Monsoon 2023. Figure 7 presents the probability density functions (PDFs) of the 15-minute maximum rainfall intensities recorded at each rain gauge for these different seasons. The Y-axis represents density, a smoothened estimate of the probability distribution of the data.*

21. In general leaf shaped or elongated watersheds generate less flood peak compared to oval or circular shaped watersheds, other factor kept constant. The current or Bindal watershed is more or less leaf shaped, I also assume that is why the peak flows a bit

lagged the peak rainfall event evident from Figure 8. Can you please discuss this issues more based on the result obtained in the current study?

Thank you for your insightful observation on the influence of watershed shape on flood peak characteristics. As you pointed out, the Bindal watershed exhibits a leaf-shaped or elongated form, which typically results in a delayed and reduced flood peak compared to more compact, oval, or circular watersheds.

In our study, the hydrograph analysis (Figure 8) indeed shows a noticeable lag between peak rainfall and peak flow, consistent with the hydrological response of elongated basins. The longer flow paths and varied travel times within such watershed geometry contribute to this lag, attenuating the flood peak. Furthermore, this shape may influence runoff concentration times and flood wave propagation, which aligns with our observations.

**Reference added:**

Venkatesh, K., Krakauer, N. Y., Sharifi, E., & Ramesh, H. (2020). Evaluating the performance of secondary precipitation products through statistical and hydrological modeling in a mountainous tropical basin of India. Advances in Meteorology, 2020(1), 8859185.

United Nations, Department of Economic and Social Affairs, Population Division (2019). World Urbanization Prospects: The 2018 Revision (ST/ESA/SER.A/420). New York: United Nations.

---

## Author Comment (AC2)

**RESPONSE TO COMMENTS ( Referee #1)**

**Integrating SMART principles in Flood Early Warning System Design in the Himalayas (NHESS-2025-2081)**

**Authors: Sudhanshu Dixit[1], Sumit Sen[1*], Tahmina Yasmin[2], Kieran Khamis[2], Debashish Sen[3], Wouter Buytaert[4], David M. Hannah[2]**

[1]Centre of Excellence in Disaster Mitigation and Management, Indian Institute of Technology Roorkee, Roorkee, Uttarakhand, India

[2]School of Geography, Earth & Environmental Sciences, University of Birmingham, Birmingham, UK

[3]People's Science Institute, Dehradun, India

[4]Department of Civil and Environmental Engineering, Imperial College London, London, UK

*Correspondence to sumit.sen@hy.iitr.ac.in

**Dear Editor and Reviewers,**

The authors would like to thank the reviewer for their careful review of our manuscript and for providing valuable comments and suggestions, which we found very helpful in improving the manuscript's quality. We have carefully addressed all your comments and integrated your insightful suggestions into the revised manuscript. In the subsequent detailed response, we have addressed each comment individually. Comments are written in red, and our responses follow each comment in black. All the new details added in the manuscript are highlighted as text in italics. We look forward to your positive feedback and hope you will find the revised manuscript satisfactory.

**Response to Reviewer 1**

1. The paper applies the main concepts of the SMART approach to catchment in the Himalayas, highlighting the relevant role of community participation in addressing the limitations of data-scarce regions and observation-based warning thresholds. To underline the importance of increasing population preparedness in catchments presenting short response times and urban settlements along rivers, the variability of recent hydrological events is investigated. The paper is generally well-organised, and the issue of improving the resilience against floods in catchments characterized by flash floods is worth investigation.

We sincerely thank the reviewer for encouraging comments regarding our application of the SMART approach to flood risk management in the Himalayan catchment. Your affirming remarks motivate us to continue refining and advancing our research in this important area.

2. Figure 1: The shape of the watershed does not correspond with the purple one in the inset. Please check;

Thank you for pointing out this potential source of confusion. The purple shape shown in the inset of Figure 1 represents the Dehradun district boundary, within which the smaller Bindal watershed is located. The main figure displays the detailed boundary of the Bindal watershed, which is a sub-catchment within the district. We have clarified this notation in the figure caption in the manuscript to avoid further confusion

3. Lines 140-144: The authors should include an image to help the reader understand the location of the mentioned stretches

Thank you for your valuable suggestion. The locations of these stretches are shown in Figure 3 of the manuscript, which we are further enhancing by adding additional map features based on suggestions from other reviewers.

[Figure]

We have added Figure 3 and a reference to Figure 3 in lines 140-144 to guide the reader directly to this visual for better understanding.

4. As community engagement is an essential part of the work, I encourage the authors to expand the description of the PRA exercises, further clarifying how their results could be applied within a real-time early warning 4 system. Since EWS typically rely on thresholds based on observed hydrological variables (e.g. water levels), how can community information be incorporated into such chains for prediction purposes? Moreover, it would be interesting to compare the information provided by the participants with that derived from recorded hydrological observations for some events, e.g. by representing on the same map the locations where the most critical flood issues were recorded and remembered. This will further clarify the role of community memories. Finally, since flood perception is often influenced by past flood experiences, it would be interesting to understand if the authors noticed differences in the information provided by the participants according to their age;

Thank you for your observation. We have revised lines 147-159, as follows.

*Participatory Rural Appraisal (PRA) exercises (Reddy et al., 2016), along with Focus Group Discussions (FGDs) with concerned community and ward members from the concerned flood-affected colonies, including slums along the four mentioned stretches, were conducted to gather crucial information required to develop early warning systems for the Bindal watershed. The PRA exercises included temporal analysis (such as historical transects, trend diagramming, and seasonal diagramming), spatial analysis (including socio-resource mapping), and relationship analysis (Venn diagrams and Scoring and Ranking). About 20-30 households participated in each of these exercises.*

*Historical transects documented events related to development (such as the establishment of colonies, roads, public buildings, protection walls, bridges, and sewer lines) along the selected river stretches, tracing back to the last 60 years. Past floods, along with their severity, were also noted. 2006, 2013, and 2018 were recently reported flood years resulting in significant damage. FGDs around the historical transect further helped in understanding the impacts of the above developmental activities on the nature of floods and related losses (riverbank erosion, public and private property loss, and livelihood loss). Trend analysis recorded decadal changes over a 40-year period, which helped examine changes in encroachment, settlement density, rainfall behavior, river base flows, flood frequency, and flood severity, as well as their correlations. For seasonal diagramming, a month-wise matrix was developed with the help of residents to understand seasonal changes with respect to river flow and record peak flows. The peak flows were reported for the period July to September, depending upon rainfall in the upper reaches of the Bindal watershed. The base flow of the river was reported to have decreased in all the stretches over the last four decades. However, the frequency and intensity of floods were reported to have increased during the last two decades.*

*During FGDs, residents identified the more vulnerable sites within concerned colonies and most affected households and structures based on experiences of previous floods. Bank erosion during floods resulted in damage to public and private property. Lohiya Nagar and Mehboob Colony (located on the right bank) and Dehra Khas (located on the right bank), in the second stretch were identified as the worst affected colonies during floods. These colonies are situated just after the confluence of the tributary (passing through Patel Nagar) with the main Bindal river. The other flood prone localities identified were Sanjay Colony (on the left bank) in the first stretch and Chota Bharuwala (on the right bank) in the third stretch, both subjected to bank erosion.*

*FGDs were held to understand the role of different institutions and effectiveness of rendered services. Scoring and Ranking exercise was undertaken with the participants to understand the extent of flood damages (loss of home, livestock, other assets, and livelihoods) across different occupational categories, namely services, business, labour, and unemployed. The daily wage labour class living in slums and having mud houses closer to the riverbanks, was most severely affected along the different stretches. At the household level, the losses reported were in terms of damage to the houses, loss of livestock and even family members, and assets like utensils, bicycles, television, etc.*

*The elderly people, those above 50 years, were more informative about the earlier status of the river, changes in rainfall behavior, encroachment along the river stretches, drainage and sewer systems, and occurrence of floods and related damages. The younger generation and middle-aged persons (20 –40 years old) provided information related to the role of different departments in river and flood management and were more interested in the flood EWS.*

*Water inundation levels for past flood events were marked on the social-resource maps, which later helped in validating and determining the different levels of thresholds, warning, and actions required. The participants were also encouraged to identify sites where water level sensors and rain gauges need to be installed.*

5. Line 234: In addition to what was mentioned at the previous point, please clarify the criterion adopted to join the "community engagement" with the flood alert thresholds reported in Table 2

Thank you for this important question regarding the integration of community engagement into the determination of flood alert thresholds. As described in lines 231-236 of the manuscript, water level thresholds were initially determined through a combination of rigorous statistical analysis and participatory validation.

The primary criterion for setting alert thresholds was statistical, using percentile-based methods to identify extreme flood events specifically; water levels exceeding the 99.9th percentile was flagged as significant. Flood events during the study period and corresponding water inundation levels in the selected stretches of rivers helped in deciding upon the threshold values. However, to ensure these threshold values were both meaningful and actionable at the local level, community engagement through referring to the water inundation levels of past floods identified during social-resource mapping exercises, was employed to adjust and confirm these figures. Interactions with elderly residents during field visits and concerned Municipal Councilors during stakeholders' workshop where the research findings were shared, contributed their experiential knowledge of past flood events, helping to validate and refine the thresholds so that they corresponded with observed flood impacts and perceptions.

6. Page 1, line 35: In addition to the acronym, please define SMART;

Thank you for your careful examination. *SMART refers to Shared understanding of risks, Monitoring of risks, building Awareness, Response action on Time.* For brevity, we have used the acronym in the abstract, as it should be limited to 200 words. However, in the main manuscript (Introduction Line 88-91: Therefore, the present work emphasizes adopting SMART approach (Shared understanding of risks, Monitoring of risks, building Awareness, Response action on Time) proposed by Yasmin et al. (2023), which promotes inclusiveness and a bottom-up strategy to maximize local relevance and effectiveness.) the acronym has been explained thoroughly before being used in the subsequent sections.

7. Please, make sure that the reference section actually includes all the cited papers: several works mentioned in the paper are actually missing, such as Papalexioux & Montanari, 2019 (line 42), Rentschler et al., 2022 (line 46), Gu et al., 2019 (line 60), and many others;

Thank you for bringing this to our attention. We have carefully reviewed the reference list and ensured that all papers cited in the manuscript, including Papalexioux & Montanari (2019), Rentschler et al. (2022), Gu et al. (2019), and others, are now correctly included and formatted in the reference section. We appreciate your diligence in helping us improve the completeness and accuracy of our citations.

8. Line 49, a point is missing before "The";

Thank you for pointing out this typographical error. We have corrected the sentence by adding the missing full stop before "The" at line 49 in the revised manuscript.

9. Line 57, a point is missing before "Rapid";

Thank you for pointing out the missing punctuation. We have corrected this typographical error by adding the missing period before "Rapid" at line 57 in the revised manuscript to improve clarity and readability.

10. Line 62: Please check the consistency of "reach become";

Thank you for pointing out this inconsistency. We agree that "reach" alone correctly conveys the intended meaning, and the phrase has been revised accordingly. The sentence now reads:

"*Furthermore, the global urban population growing from 13% in 1900 to 49% in 2005 is estimated to reach 60% by 2030.*" This change can be found in the revised manuscript.

11. Lines 180-181: R1, R2, R3, and R4 are actually RG1, RG2, RG3, and RG4. Please correct;

Thank you for pointing out this inconsistency. We have carefully reviewed the manuscript and corrected the abbreviations in lines 180-181 from R1, R2, R3, and R4 to the correct RG1, RG2, RG3, and RG4 throughout the text.

12 Line 383: Please check the sentence;

Thank you for your valuable comment regarding the clarity of the statements in lines 383 to 387. We agree that the original text could be more clearer. Accordingly, we have rewritten this section to improve readability and provide a more precise explanation of the analysis and figure presentation. The revised text now reads as follows:

"*Rainfall and water level data were analyzed to understand watershed dynamics and their relationship with flood warnings. The peak rainfall intensity at 15-minute intervals was calculated for each rain gauge during three distinct periods: (a) Monsoon 2022, (b) Non-monsoon 2023, and (c) Monsoon 2023. Figure 7 presents the probability density functions (PDFs) of the 15-minute maximum rainfall intensities recorded at each rain gauge for these*

*different seasons. The Y-axis represents density, a smoothened estimate of the probability distribution of the data.*

13. Reference section: Some references are listed twice. Please check.

Thank you for noticing this issue. We have carefully reviewed and cleaned the reference section to remove any duplicate entries, ensuring that each reference appears only once. The reference list in the revised manuscript is now updated.

---

## Author Comment (AC3)

**RESPONSE TO COMMENTS (Referee #2)**

**Integrating SMART principles in Flood Early Warning System Design in the Himalayas (NHESS-2025-2081)**

**Authors: Sudhanshu Dixit[1], Sumit Sen[1*], Tahmina Yasmin[2], Kieran Khamis[2], Debashish Sen[3], Wouter Buytaert[4], David M. Hannah[2]**

[1]Centre of Excellence in Disaster Mitigation and Management, Indian Institute of Technology Roorkee, Roorkee, Uttarakhand, India

[2]School of Geography, Earth & Environmental Sciences, University of Birmingham, Birmingham, UK

[3]People's Science Institute, Dehradun, India

[4]Department of Civil and Environmental Engineering, Imperial College London, London, UK

*Correspondence to sumit.sen@hy.iitr.ac.in

**Dear Editor and Reviewer,**

The authors would like to thank the reviewer for the careful review of our manuscript and for providing us with their valuable comments and suggestions, which we have found very helpful in improving the quality of the manuscript. We have carefully addressed all your comments and integrated your insightful suggestions into the revised manuscript. In the subsequent detailed response, we have addressed each comment individually. Comments are written in red, and our responses follow each comment in black. All the new details added in the manuscript are highlighted as text in italics. For your reference, the sources cited in our responses can be found in the references section on the last page of this document. We look forward to your positive feedback and hope you will find the revised manuscript satisfactory.

**Response to Reviewer 2**

1. The study presents an integrated approach to the design of early warning systems for flash floods in both urban and mountainous environments, combining real-time monitoring technologies with the active involvement of local communities.
   The authors implement a high-resolution hydrometeorological network, based on LiDAR sensors and advanced measurement instruments, aimed at providing a detailed characterization of the rainfall and hydrological regime of the study basin. The data collection and analysis highlight a marked spatial variability in precipitation (up to more than 180 mm between stations only a few kilometers apart), a crucial factor for forecasting localized flood events.
   The SMART model constitutes the conceptual core of the study and, although not clearly defined, is described as a dynamic and adaptive system. The comparison with global reanalysis (ERA5) and satellite (GPM) datasets shows that these sources fail to

adequately capture the complexity of precipitation patterns in mountainous areas. In contrast, the SMART approach proposed here, based on real-time local data, integrates basin dynamics and adapts alert thresholds in a context-specific manner. A key methodological feature is the definition of thresholds based on percentiles of water level data, validated through community participation, which makes the system more flexible and responsive to actual environmental conditions. Although the work represents an original contribution to the literature on early warning systems for extreme events, it also presents some limitations, several of which are acknowledged by the authors themselves.

Thank you very much for your thoughtful and thorough assessment of our study. We appreciate your recognition of the integrated approach to early warning system design, which combines state-of-the-art monitoring technologies with active community participation across an urban, mountainous, and Himalayan catchment.

We are pleased that you found the implementation of the high-resolution hydrometeorological network and characterization of rainfall regime valuable, as well as the investigation of spatial precipitation variability, which is one of the critical factors in flash flood warning. As highlighted, the comparison with global datasets (ERA5 and GPM) shows the need for locally adaptive systems, and our SMART model aims to address this through real-time data integration and dynamic, context-specific alerting.

We appreciate your acknowledgment of our percentile-based thresholding methodology, which we validated through community involvement, as a flexible and responsive approach in data-scarce regions. As you noted, while our study makes an original contribution, we remain committed to further refinement and transparency. We have acknowledged limitations in greater detail in the revised version.
Thank you again for your constructive review and encouragement, which help strengthen the study's relevance and impact.

2. Limited spatial and temporal extent – The study focuses on a single small catchment (Bindal, 44.4 km$^2$) and a relatively short observation period of approximately one year (September 2022 – August 2023). This timeframe is insufficient to robustly assess the system's performance with respect to interannual variability or its ability to capture rare extreme events. The authors themselves acknowledge that the hydrological response of the basin is strongly influenced by local-scale factors, underscoring the need for further testing and validation across broader spatial and temporal scales.

Thank you for bringing this important issue related to spatial and temporal scope to our attention. The Bindal catchment (44.4 km$^2$) is representative of various Himalayan urban and semi-urban areas, which commonly range between 30 and 100 km$^2$, making our findings relevant to similar contexts (Bharti et al., 2020; Mani et al., 2025). Additionally, we would like to emphasize that conducting sustained hydrometeorological monitoring in the Himalayan region is logistically challenging due to steep terrain, limited accessibility, harsh climatic conditions, and frequent sensor damage during extreme

events. These factors contribute to a broader regional challenge, namely a severe scarcity of continuous, high-resolution datasets, which limits rigorous long-term analysis. Concerning temporal coverage, we acknowledge that the original observation period of approximately one year (September 2022 – August 2023) is limited for capturing longer-term interannual variability and rare extreme events. To address this, we have extended our analysis to include additional rainfall data from August 2024 to October 2025, which reveals significant rainfall variability, with differences ranging from 50 mm to 200 mm between adjacent rain gauges within a single week, as shown in the figure below. This extended temporal dataset, included in the supplementary materials in bar plot form, provides further evidence of the system's spatially variable events within the watershed, indicating that the spatial variability remains consistent.

[Figure]

However, due to funding constraints, deploying and validating the SMART approach in additional watersheds remains challenging. *Nonetheless, the conceptual strength of SMART lies in its adaptive, community-engaged framework that captures local hydrological dynamics and integrates participatory validation, enabling it to provide actionable warnings tailored to catchment-specific conditions.* We have emphasized these points and limitations in the revised manuscript and highlighted the necessity for future broader spatial and temporal validation studies.

**3.** Lack of operational validation – The study does not include a verification of the system under real operational conditions, nor does it present application-based simulations for future events. Moreover, no performance metrics are provided — such as lead time, false alarm rate, or the accuracy of dynamic thresholds — which are essential for a quantitative assessment of the system's effectiveness.

Thank you for highlighting the absence of operational validation and detailed performance metrics in our study. We acknowledge that metrics such as lead time, false alarm rate, and threshold accuracy are crucial for assessing the effectiveness of early warning systems under real conditions. Due to limited funding and resource constraints, this research was designed as a conceptual and demonstration study, focusing on the technical feasibility and local relevance of our approach in a data-scarce Himalayan context.

At this stage, we have not yet conducted real-event operational deployments or simulated future events. However, our results provide encouraging indications regarding the potential of our proposed framework to be made operational. Specifically, we observed that the lag time between rainfall peak and water level peak varies between 15 to 45 minutes, which in future operational scenarios could be harnessed as actionable lead time for flood warnings. We agree that operational validation, which includes simulations and robust quantitative metrics, is the key next step, and we have emphasized this in the revised discussion of future work.

4. Dependence on percentile thresholds – The system relies on statistical thresholds derived from percentiles of water level data; however, the study does not provide an in-depth analysis of the model's sensitivity to variations in these thresholds, nor does it address its ability to manage exceptional or out-of-scale events.

Thank you for raising the important issue regarding the system's reliance on percentile-based water level thresholds, and your request for more detail on sensitivity and robustness to exceptional events. *In our study, the thresholds were defined through both a statistical approach and community validation, using a comprehensive dataset of water levels spanning a five-minute interval from April 2022 to May 2024, comprising over 200,000 water level data points at a single location. This extensive record enabled us to capture a wide range of hydrological conditions and extract major flood events. These statistically derived thresholds were subsequently reviewed and verified by community members based on both historical experience and recent observations.*

It is recognized that percentile-based thresholding is a common and practical approach, particularly in settings with limited resources, and has been applied in recent literature for early warning systems (Jiang et al., 2023). *One positive aspect of our method is its adaptability; as more data are collected over time, the thresholds can be recalibrated, steadily enhancing their robustness to changing hydrological conditions.*

5. Non-formalized community involvement – Although the study places significant emphasis on local community participation, it does not present a structured and replicable methodology for systematically integrating community knowledge into the decision-making process. Furthermore, operational details on how surveys and consultations were conducted are lacking. The overall effectiveness of the system largely depends on the level of community engagement and technical capacity — factors that may limit its transferability to other socio-cultural contexts.

The operation and maintenance of EWS can only be ensured through the active participation of the concerned residents. The research team first conducted a transect survey of the entire Binal watershed in consultation with the residents to identify flood prone areas along the river. This helped in identifying four flood prone stretches in the lower reach of the Binal watershed, based on nature and the extent of damage from previous flood episodes. Thereafter, the team visited the affected families in these flood-prone areas, requesting their participation in PRA and FGD exercises to develop an effective EWS based on their inputs, which would benefit them in the future.

During the PRAs, all sections of the community, including men, women, youth, and elderly people, as well as those with different livelihood sources, were encouraged to participate. Before engaging them in the activities, it was essential to create an atmosphere that fostered active participation. The community leaders played a critical role in personally mobilizing the concerned families. The place and time for interaction were determined based on the suitability of the residents for each of the concerned stretches.

In each of the FGDs, 20-30 residents shared their experiences about flooding, its causes, and related damages. They indicated water levels of past flood episodes, which later helped in finalizing the threshold values. At the end of these interactions, the households concerned felt the need for an EWS as it would save the lives of people, their livestock, and their assets. As a result, they took an active interest in suggesting sites for the installation of rain gauges and water level sensors. They also assured the research team of taking care of the installed devices. Once the devices were installed, the research team regularly visited the different stretches, jointly observing the water levels and flood damage during the monsoon period, which helped gain confidence in the communities concerned and in finalizing of the flood threshold.

6. Absence of predictive hydrological or hydraulic modelling – The study focuses primarily on monitoring activities and descriptive data analysis, but does not incorporate physical or predictive runoff models capable of simulating future scenarios or assessing the impacts of anthropogenic changes and climatic variations. The authors themselves acknowledge the need to integrate hydrological modelling components to enhance the operational effectiveness of the early warning system.

Thank you for bringing this important point to our attention regarding the absence of predictive hydrological or hydraulic modeling components in our work. As acknowledged in the manuscript, the present study provides a framework that demonstrates the feasibility and value of a high-resolution, community-engaged monitoring and early warning system in a data-scarce Himalayan setting. We have installed four telemetry rain gauges with a temporal resolution of 15 minutes and three water level sensors collecting data every 5 minutes. This community-engaged framework captures local hydrological dynamics and integrates participatory validation, enabling it to provide actionable warnings tailored to watershed-specific conditions. Given current resource and funding constraints, our efforts were directed at establishing proof-of-concept for real-time monitoring and participatory threshold validation, rather than on the development and integration of comprehensive predictive models at this stage. We fully agree that integrating process-based, physics-driven hydrological and hydraulic models, which enable the simulation of future scenarios and climatic variations, will be a critical future step in enhancing operational effectiveness and early warning capabilities. This is highlighted as a clear priority in the future scope in the revised manuscript.

**Specific Comments:**

7. Abstract: It should be clarified more explicitly whether the implemented procedure is intended for nowcasting or forecasting purposes. Although the term forecasting is mentioned in line 101 of the introduction, the use of real-time precipitation and runoff monitoring tools might suggest a nowcasting application, which is generally impractical for a catchment of such limited size. Therefore, the operational objective

of the system should be specified more clearly. It is presumed that the real-time data were primarily used to calibrate a forecasting model.

Thank you for your valuable feedback. We recognize the importance of distinguishing between nowcasting and forecasting in hydrometeorological applications. The Bindal watershed exhibits a range of response times from 15 minutes to 2 hours and 30 minutes. Specifically, this response time varies from 15 to 45 minutes during heavy and very heavy rainfall events. Given these rapid dynamics and short lead times, the system, is best categorized as a nowcasting system. While the introduction previously mentioned "forecasting," we have revised the manuscript and abstract to nowcasting. The real-time data are primarily used to trigger and refine nowcast-based alerts, rather than calibrating or operating a forecasting model.

8. Figure 2: It would be advisable to clarify whether the four key components hold the same level of importance within the methodological framework. The graphical representation appears to suggest equal weighting, but a brief explanation in the text would help to better understand any hierarchical or functional relationships among them.

Thank you for your suggestion regarding clarification of the methodological framework illustrated in Figure 2. We confirm that all four key components hold equal weight within our approach and are closely interdependent. To address your comment, we have added a sentence in the manuscript at the end of the methodology description (lines 131–134), stating "*These four key components are equally important and are operationally interlinked, forming a sequential and interdependent framework essential for the robustness and effectiveness of the early warning system (Figure 2)."*

9. Lines 140–145: Including a geographic reference figure in this section would enhance the spatial understanding of the study area and clarify the subdivision of the river segments. At present, Figure 3 provides only partial information, and its placement far from the relevant text reduces its immediacy and readability.

Thank you for this helpful suggestion for improving spatial clarity in the manuscript. We agree that including a geographic reference figure closer to lines 140–145 would enhance the understanding of the study area. In response, we have updated Figure 3 to more comprehensively illustrate the locations and boundaries of the relevant river stretches and reposition the figure in section 3.1 nearer to the corresponding text section to improve immediacy and readability. A direct reference to Figure 3 will also be added in lines 140–145 to guide readers efficiently to the spatial context.

The updated figure can be found in the revised manuscript (Figure 3).

10. Paragraph 3.1 – Community Interaction: In this section, it is not clearly explained how community interaction contributed to the definition or adjustment of the thresholds. Including examples of the questions asked to participants, along with a map of the

most vulnerable areas and an explanation of how these maps were produced, would make the methodology more transparent and easier for the reader to understand.

Thank you for this critical comment regarding the role of community interaction in defining and adjusting threshold values. We have revised section 3.1 to provide a more precise and detailed explanation of this process. Participatory exercises, stakeholders' workshops and structured household interviews were used to specifically engage members of the local communities from the flood prone areas for determining the threshold values. Participants during PRAs and FGDs, held separately for the four identified flood prone stretches of the river, reported their historical experiences and observations of different water levels corresponding to different flood episodes from their memories. They were encouraged to mark the water levels on the respective social-resource maps prepared by them. Figures 3a) and 3b) have now been added, indicating the most vulnerable colonies generated based on the above mapping exercises and community interactions. This qualitative knowledge was used to validate the statistical percentile-based thresholds derived from the monitoring data, allowing us to fine-tune and adjust these thresholds to better reflect local flood perceptions and realities. The process of participatory validation ensured that the alert levels were not only statistically robust but also meaningful and actionable for the affected populations, thereby enhancing the overall effectiveness and acceptance of the early warning system.

11. Lines 175–180: It is recommended to revise Figure 1 by moving part 1a to this section, in order to immediately show the location of the sensors and facilitate comprehension, avoiding the need for the reader to flip back several pages. It would also be useful to indicate the distance of the discharge measurement sensors along the river channel.

> Thank you for the insightful suggestion. To improve readability and spatial comprehension, we have added a new figure (3a) to Section 3.1, located around lines 175–180, which allows readers to immediately view the sensor locations without needing to refer back.

[Figure]

Additionally, we have included a new supplementary table providing detailed information on the water level sensors along the river channel.

| Sensor | Name | Latitudes | longitudes |
|---|---|---|---|
| **Rainguage1** | RG1 | 30.39833333 | 78.09541667 |
| **Rainguage2** | RG2 | 30.35973333 | 78.05485 |
| **Rainguage3** | RG3 | 30.34092778 | 78.02178889 |
| **Rainguage4** | RG4 | 30.33598611 | 78.04876389 |
| **Waterlevel1** | WL1 | 30.335517 | 78.038947 |
| **Waterlevel2** | WL2 | 30.36703333 | 78.06797222 |
| **Waterlevel3** | WL3 | 30.31888889 | 78.01607222 |

.**Table ST2: Location details and geographic coordinates of the hydrological monitoring network sensors in the Bindal watershed. Latitudes and longitudes are provided in decimal degrees.**

| | RG1 | RG2 | RG3 | RG4 | WL1 | WL2 | WL3 |
|---|---|---|---|---|---|---|---|
| **RG1** | 0 | | | | | | |
| **RG2** | 6.48 | 0 | | | | | |
| **RG3** | 9.38 | 4.49 | 0 | | | | |
| **RG4** | 6.86 | 3.58 | 2.96 | 0 | | | |
| **WL1** | 7.55 | 2.84 | 1.73 | 0.85 | 0 | | |
| **WL2** | 5.39 | 2.14 | 6.97 | 4.67 | 4.67 | 0 | |
| **WL3** | 11.16 | 6.65 | 2.55 | 3.78 | 3.32 | 7.35 | 0 |

**Table ST3: Inter-sensor distance matrix for the Bindal watershed hydrological monitoring network {km}.**

12. Lines 188–193: It may be more effective to place the entire Figure 1 in this section to ensure consistency between the text and the illustrations. Alternatively, a new figure could be added in Chapter 2, specifically dedicated to the description of the study area.

Thank you for your suggestion. As mentioned in our response to comment no. 11, we have added a new Figure 3a in section 3.1 to enhance readability.

13. Line 240 (Table 2): It is advisable to adjust the table background, as the first row is difficult to read due to the low contrast between the text and the background color. Increasing the contrast would significantly improve readability.

Thank you for your valuable observation regarding the readability of Table 2. We have removed the table background and text colors in the revised manuscript to increase contrast. Same table is shown below for your reference.

| Type of alert | Threshold | Action |
|---|---|---|
| Warning | 99.99 percentile of Water level | Flood-like situation: Evacuate. |
| Advisory | 99.9 percentile of Water level | Flood-like situation: Stay away from banks |

| Watch | 99.5 percentile of Water level | Stay alert |
| Information statement | 99 percentile of Water level | No action required |
| Cancellation | Below 99 percentile of Water level | Safety confirmed |

14. Some comments from the interactive discussion are not repeated here, but I fully agree with them and suggest the authors address those points as well.

Thank you for your positive feedback and for emphasizing the importance of addressing all comments raised during the interactive discussion. We have carefully reviewed all such points and incorporated corresponding revisions and clarifications throughout the manuscript and response letter. We appreciate your thorough evaluation and support, which have greatly strengthened the quality and clarity of our work.

**References added:**

Bharti, N., Khandekar, N., Sengupta, P., Bhadwal, S., & Kochhar, I. (2020). Dynamics of urban water supply management of two Himalayan towns in India. Water Policy, 22(S1), 65-89.

Mani, A., Kumari, M., Badola, R. (2025). "Urban Watershed Management in the Doon Valley: A Geospatial Assessment of Himalayan Watersheds." Journal of Landscape Ecology.